# Ligand-switchable nanoparticles resembling viral surface for sequential drug delivery and improved oral insulin therapy

Tiantian Yang[1,2], Aohua Wang[1,2], Di Nie[1,2], Weiwei Fan[1,2], Xiaohe Jiang[1,2], Miaorong Yu[1,2], Shiyan Guo[1], Chunliu Zhu[1], Gang Wei [3] ✉ & Yong Gan [1,2,4] ✉

Mutual interference between surface ligands on multifunctional nanoparticles remains a significant obstacle to achieving optimal drug-delivery efficacy. Here, we develop ligand-switchable nanoparticles which resemble viral unique surfaces, enabling them to fully display diverse functions. The nanoparticles are modified with a pH-responsive stretchable cell-penetrating peptide (Pep) and a liver-targeting moiety (Gal) (Pep/Gal-PNPs). Once orally administered, the acidic environments trigger the extension of Pep from surface in a virus-like manner, enabling Pep/Gal-PNPs to traverse intestinal barriers efficiently. Subsequently, Gal is exposed by Pep folding at physiological pH, thereby allowing the specific targeting of Pep/Gal-PNPs to the liver. As a proof-of-concept, insulin-loaded Pep/Gal-PNPs are fabricated which exhibit effective intestinal absorption and excellent hepatic deposition of insulin. Crucially, Pep/Gal-PNPs increase hepatic glycogen production by 7.2-fold, contributing to the maintenance of glucose homeostasis for effective diabetes management. Overall, this study provides a promising approach to achieving full potential of diverse ligands on multifunctional nanoparticles.

The surface functionalization of nanoparticles with various types of ligands with chemical or biological activity is a powerful tool for efficient drug delivery. These multifunctional nanoparticles have significant potential for overcoming complex physiological barriers and increasing the targeting efficiency of encapsulated drugs. This task is challenging to accomplish with a single ligand[1,2]. For example, the covalent modification with tripeptide Arg-Gly-Asp (RGD) and transferrin (Tf) ligands enables nanoparticles to transit across tumor vascular barriers and enhance the cellular uptake of drugs, respectively, leading to better antitumor efficacy[3]. However, mutual interference (e.g., steric hindrance and electrostatic interactions) between diverse surface ligands would sterically hinder binding to receptors, ultimately compromising their functions[4,5] and resulting in low efficacy of multifunctional nanoparticles[6,7]. Therefore, there remains a significant

challenge related to the surface multifunctionalization of nanoparticles, which substantially determines the efficiency of in vivo drug delivery.

Several strategies have emerged in recent decades to address the challenge by improving synergism between diverse ligands or controlling the presentation of specific ligands. One such strategy is to optimize the relative length, ratio, and density of dual-targeting ligands on nanoparticles by screening for the optimal formulation[8,9]. An alternative strategy involves utilizing enzyme-responsive linkers to anchor one ligand on nanoparticles and cleaving the linker by specific enzymes to expose another ligand, enabling it to exert functions at the target site[10,11]. In addition, polyhistidine, a pH-sensitive molecular chain actuator, has also been applied to selectively expose conjugated functional moieties in response to acidic environments[12,13]. Although

[1]State Key Laboratory of Drug Research, Shanghai Institute of Materia Medica, Chinese Academy of Sciences, Shanghai 201203, China. [2]University of Chinese Academy of Sciences, Beijing 100049, China. [3]Key Laboratory of Smart Drug Delivery, Ministry of Education, Department of Pharmaceutics, School of Pharmacy, Fudan University, Shanghai 201203, China. [4]NMPA Key Laboratory for Quality Research and Evaluation of Pharmaceutical Excipients, National Institutes for Food and Drug Control, Beijing 100050, China. ✉e-mail: weigang@shmu.edu.cn; ygan@simm.ac.cn

these strategies have succeeded in improving the targeting efficiency of nanoparticles, there still remains a major bottleneck in fully realizing the multiple functions of various ligand-modified nanoparticles.

Herein, to minimize mutual interference between ligands and improve the performance of multifunctional nanoparticles, we develop ligand-switchable poly (lactic-co-glycolic acid) (PLGA) nanoparticles (PNP) modified with a pH-triggered stretchable cell-penetrating peptide (Pep) and a hepatic targeting moiety (galactose, Gal) (Pep/Gal-PNPs). The Pep mimics spike proteins on the viral surface (i.e., hemagglutinin on influenza A virus[14] and coronavirus spike proteins[15]) that experience conformational changes in response to pH, which helps to avoid mutual interference between diverse functionalities. After the oral administration of Pep/Gal-PNPs, Pep adopts a stretched conformation and extends from the surface in acidic environments, mediating the efficient traversal of intestinal barriers. Subsequently, upon entering systemic circulation, Gal is exposed on the surface after Pep folds at physiological pH, thereby specifically guiding Pep/Gal-PNPs to the liver (Fig. 1). In this regard, the Pep/Gal-PNPs resemble unique viral surface features that modulate conformations of surface proteins to enable sequential display of diverse functions[16]. Since oral insulin delivery requires the stepwise processes of traversing intestinal barriers and targeting the liver to restore the liver–periphery insulin gradient and correct glucose metabolism defects in the context of diabetes[17], as a proof-of-concept, we apply the Pep/Gal-PNPs for oral insulin therapy in this study. The in vivo results indicate that insulin-loaded Pep/Gal-PNPs not only elicit significant hypoglycemic effects but also promote hepatic glucose sequestration and glycogen storage in diabetic rats, which show similar glucose utilization as normal rats.

In summary, the Pep/Gal-PNPs could fully exert diverse functions of dual surface ligands by modulating the conformations of Pep ligands, resembling virus in unique surface features, to avoid their mutual interferences and ultimately improving oral insulin therapy. Furthermore, this work presents a promising alternative to recent attempts at the surface multifunctionalization of nanocarriers, which is anticipated to be applied for a broad range of scenarios, such as oral delivery of biomacromolecules and targeting the delivery of antitumor drugs, playing a significant role in improving their in vivo therapeutic efficacy.

## Results

### Preparation and characterization of Pep and PLGA-based functional polymers

To prepare the multifunctional nanoparticles, the Pep and PLGA-conjugated functional polymers, including PLGA-Pep and PLGA-PEG-Gal, were synthesized and characterized. The Pep $R_6G_5(HE)_{10}$, which consists of arginine (R), glycine (G), and histidine–glutamic acid (HE) repeats, is sensitive to environmental pH[18]. In acidic environments (pH <7), the Pep was in the open state with a theoretical length of 10.85 nm, whereas it switched to a closed state under physiological conditions (pH ~7.4), with an estimated length of 7.00–7.88 nm depending on the folding pattern (Fig. 2a), according to previous studies[19]. The synthesized Pep was validated by mass spectrometry (Fig. 2b) and proton nuclear magnetic resonance (¹H NMR) (Supplementary Fig. 1). Then, the pH-triggered conformational changes of Pep were confirmed by circular dichroism (CD). The spectra revealed that Pep adopted a random coil conformation, with the minimum absorption at 198 nm,

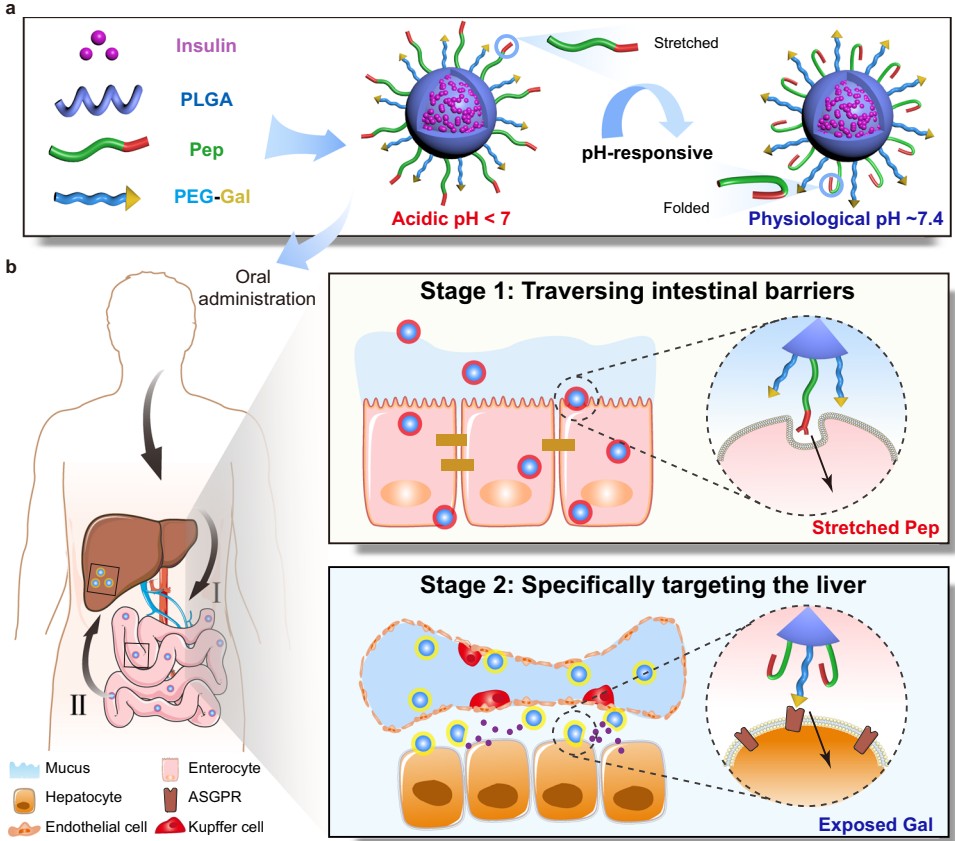

**Fig. 1 | Schematic illustrations of composition and mechanism of Pep/Gal-PNPs for oral insulin delivery. a** The construction of virus surface-inspired ligand-switchable nanoparticles (Pep/Gal-PNPs) modified with both a pH-triggered stretchable cell-penetrating peptide (Pep) and a hepatic targeting moiety (galactose, Gal). **b** After oral administration, Pep adopts a stretched conformation in response to the acidic environment in the intestine and mediates efficient Pep/Gal-PNPs transport across intestinal barriers. Subsequently, Gal is exposed on the surface as Pep folds at physiological pH in circulation and specifically guides Pep/Gal-PNPs to the liver.

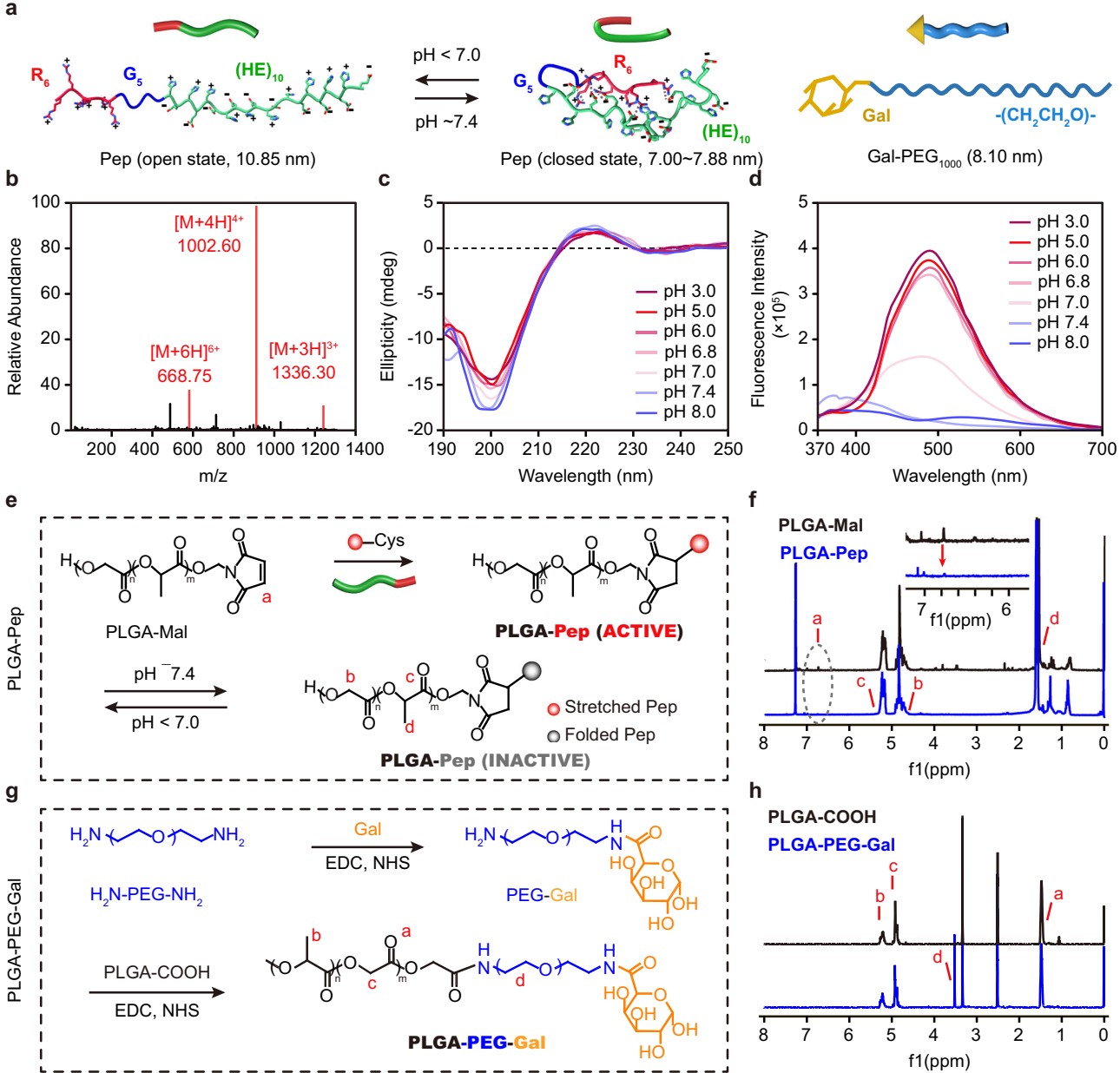

**Fig. 2 | Synthesis and characterization of functional polymers. a** Schematic illustration of pH-responsive stretchable cell-penetrating peptide (Pep) and polyethylene glycol-galactose (PEG-Gal) polymers and their theoretical lengths. Open- and closed-state models of Pep under different pH conditions. H: histidine; E: glutamic acid; G: glycine; R: arginine. **b** Mass spectrum of Pep. **c** Circular dichroism spectra of Pep under different pH conditions. **d** Emission spectra of FRET pair-labeled Pep under different pH conditions. **e** The synthetic route to PLGA-Pep polymers. **f** $^1$H NMR spectra of PLGA-Mal and PLGA-Pep polymers. Characteristic peaks are assigned according to the labels in panel (**e**). **g** The synthetic route to PLGA-PEG-Gal polymers. **h** $^1$H NMR spectra of PLGA-COOH and PLGA-PEG-Gal polymers. Characteristic peaks are assigned according to the labels in panel (**g**).

and underwent noticeable changes as the pH increased (Fig. 2c). The ratio of β-sheet in the secondary structure of Pep was estimated to increase from 11.9% to 26.5% when pH increased from 6.8 to 7.4 by analyzing the CD spectra using Spectra Manager software. As the β-sheet is the most common structure in the folding pattern of proteins and peptides[20], the results indicated that Pep folded at physiological pH.

Furthermore, fluorescence resonance energy transfer (FRET) technique with the Edans (fluorophore) and Dabcyl (quencher) pair was used to confirm the structural changes of Pep in response to pH. There was a significant overlap between the Edans emission spectrum and the Dabcyl absorption spectrum in the pH range of 3.0–8.0 (Supplementary Fig. 2), demonstrating that Dabcyl could absorb the fluorescence emitted by Edans. Subsequently, the Edans and Dabcyl were conjugated to the amino acid side groups of N- and C-termini of Pep, respectively; the fluorescence intensity of Edans decreased sharply when the pH increased from 6.8 to 7.4 (Fig. 2d), indicating that the two ends of Pep became closer upon adopting a folded conformation at physiological pH. These results suggested that Pep underwent pH-responsive conformational changes similar to viral spike proteins, with a stretched conformation at acidic pH and a folded one at physiological pH. Moreover, we also studied changes in the activity of Pep in response to pH, and results revealed that Pep exhibited a potent hemolytic effect at pH <7.0 but was inactive at physiological pH, indicating that Pep was activated in a pH-dependent manner (Supplementary Fig. 3).

Subsequently, PLGA-conjugated functional polymers were further synthesized. A cysteine was added to the C-termini of Pep to enable conjugation with maleimide-capped PLGA (PLGA-Mal), thus creating PLGA-Pep polymers (Fig. 2e). The maleimide peak at 6.8 ppm in the $^1$H NMR spectrum disappeared after conjugation with Pep (Fig. 2f), indicating the successful synthesis of PLGA-Pep polymers. To selectively expose the other functional ligand (Gal) when Pep folded, polyethylene glycol (PEG, MW 1.0 kDa) with an estimated length of 8.10 nm[8] was used as the linker. The PEG chain was first conjugated with Gal and then coupled with PLGA-COOH through an amidation reaction to obtain PLGA-PEG-Gal polymers (Fig. 2g). The intermediate product, PEG-Gal polymers, was monitored via $^1$H NMR (Supplementary Fig. 4), and the structure of PLGA-PEG-Gal was confirmed by the representative PEG methylene signal at 3.6 ppm in the final spectrum (Fig. 2h).

## Preparation and characterization of Pep/Gal-PNPs

The PLGA nanoparticles (PNPs) functionalized with various ligands were prepared through a double emulsion and solvent evaporation method, as previously reported[21]. The modification rates for the Pep and Gal ligands on nanoparticles were both approximately 5% (Supplementary Table 1). The switchable surface properties of Pep/Gal-PNPs were first investigated by dynamic light scattering (DLS) to determine the hydrodynamic diameter and zeta potential at different pH. The results showed that the size of PNPs increased significantly after ligand modification (Fig. 3a). When pH increased from 6.8 to 7.4, the diameter of Pep-modified PNPs (Pep-PNPs) decreased by approximately 20 nm (Fig. 3a), suggesting that Pep folded at physiological pH. Although the size of Pep/Gal-PNPs decreased only slightly as the pH increased due to the presence of PEG-Gal on the surface, the observed differences in size under different pH conditions revealed the switchable nature of the dual surface ligands (Fig. 3a). The zeta potential was determined as another characterization of Pep/Gal-PNP

surface properties. For Pep-PNPs and Pep/Gal-PNPs, the zeta potential sharply transitioned from positive to negative when the pH increased from 6.8 to 7.4 (Fig. 3b). By contrast, PNPs remained at approximately −35 mV regardless of pH (Fig. 3b). As a previous study showed that the cationic arginine in Pep could be neutralized by anionic glutamic acid at pH 7.4[22], we hypothesized that changes in electrostatic interactions between the amino acids of Pep might underlie the charge reversal of nanoparticles. Moreover, the morphologies of PNPs, Pep-PNPs, and Pep/Gal-PNPs were observed by cryogenic transmission electron microscopy (cryo-TEM). All the nanoparticles showed spherical morphologies with uniform size (Fig. 3c).

To further elucidate the switchable nature of the surface ligands, the ligand corona around the nanoparticles was directly observed by atomic force microscopy (AFM). We detected the changes in individual immobilized nanoparticles under different pH conditions by scanning the same position on the silica substrate and found that the diameter of Pep-PNPs was smaller at pH 7.4 than at pH 6.8 (Fig. 3d, top row); specifically, the thickness of the Pep corona decreased by approximately 8 nm (from 25.86 ± 2.54 nm to 17.08 ± 0.60 nm) as the pH increased. By contrast, the ligand corona around Pep/Gal-PNPs decreased slightly over the same pH shift (Fig. 3d, bottom row), which was consistent with the DLS results. Moreover, no significant changes were detected in the size of Gal ligand-modified nanoparticles (Gal-PNPs) under different pH conditions; the thickness of PEG-Gal corona around nanoparticles remained in the range of 21–23 nm (Supplementary Fig. 5). Therefore, the AFM results confirmed that Pep underwent pH-responsive structural changes, suggesting that multifunctional Pep/Gal-PNPs featured switchable ligands with similar surface properties as viruses. Furthermore, we investigated the activity of Pep after decoration on Pep/Gal-PNPs, and results indicated it still retained the pH-dependent hemolytic effect (Supplementary Fig. 6). In contrast,

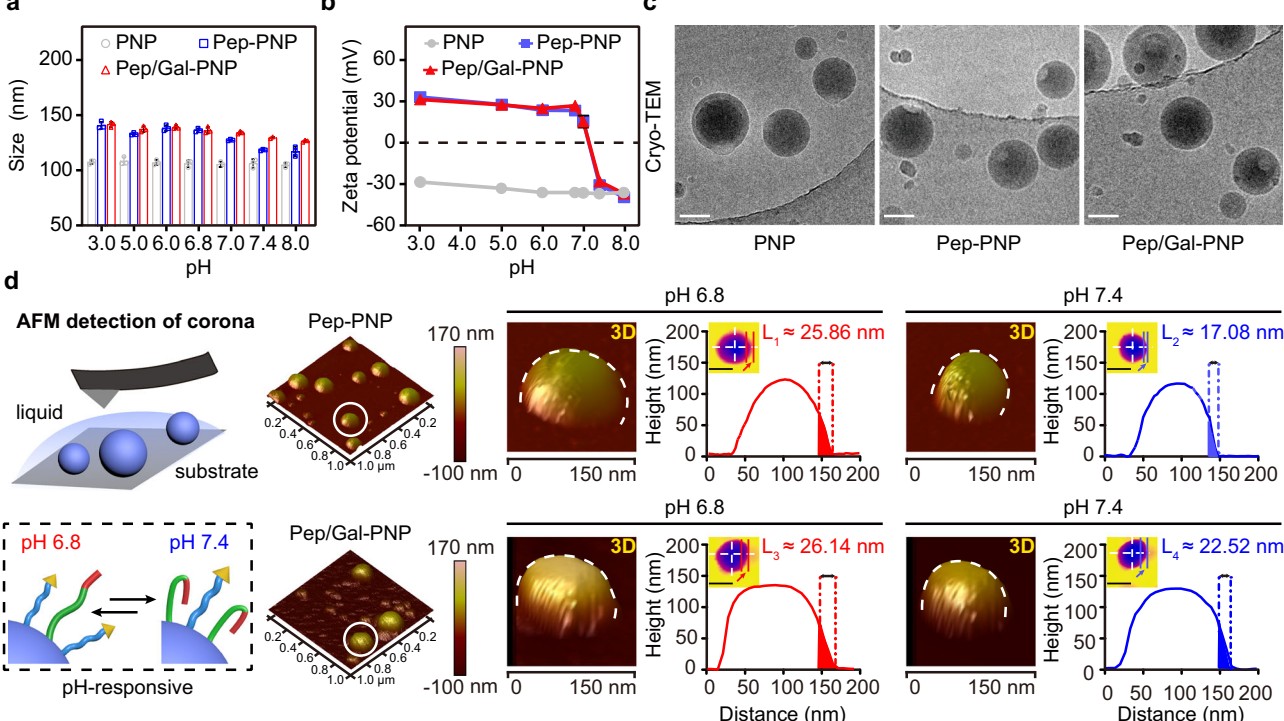

**Fig. 3 | Synthesis and characterization of Pep/Gal-PNPs. a, b** The size and zeta potential of nanoparticles under different pH conditions. Data are presented as the mean ± SD ($n$ = 3 biologically independent experiments). **c** Cryo-TEM images of nanoparticles. **d** Atomic force microscopy (AFM) analysis of nanoparticles under fluid conditions. 3D modeling images, height maps, and height profiles of Pep-PNPs (top row) and Pep/Gal-PNPs (bottom row) at pH 6.8 and 7.4. Scale bar: height map, 100 nm. The thickness of ligand corona around the nanoparticles (as indicated by the arrows in the height map) was analyzed using NanoScope Analysis software. Representative images are presented, and the data are means ± SD ($n$ = 3 biologically independent experiments). Source data are provided as a Source Data file.

PNPs exhibited relatively low hemolysis at all pH values (Supplementary Fig. 6).

Insulin, a protein drug widely used for the treatment of diabetes[23], was selected as the model drug in this study. The insulin entrapment efficiency and loading capacity of Pep/Gal-PNPs were determined to be 48.1% and 7.9%, respectively (Supplementary Table 1). Then, to investigate Pep/Gal-PNP stability after oral administration, the nanoparticles were incubated in PBS, simulated gastric fluid (SGF), and simulated intestinal fluid (SIF) with digestive enzymes for 4 h, respectively. The results showed that the relative size and dispersity of Pep/Gal-PNPs were not significantly different in SGF and SIF compared with that in PBS (Supplementary Fig. 7). Moreover, Pep/Gal-PNPs showed sustained insulin release in vitro, with approximately 23% of insulin released in 4 h (Supplementary Fig. 8a), and CD results demonstrated that the released insulin still retained a similar structure as native insulin (Supplementary Fig. 8b). Overall, these results indicated that Pep/Gal-PNPs could remain stable in the harsh gastrointestinal environment without the premature release of encapsulated insulin.

## Overcoming the mucus barrier

After being subjected to the harsh gastrointestinal environment, the next barrier encountered by Pep/Gal-PNPs is the mucus lining of the intestines, which must be crossed to reach the apical side of the intestinal epithelium[24,25]. Therefore, the ability of nanoparticles to penetrate mucus was investigated using HT29-MTX-E12 (E12) cells that secrete abundant mucus. Pep/Gal-PNPs exhibited strong fluorescence at the lower layer of mucus at pH 6.8 and 7.4 (Supplementary Fig. 9), suggesting the efficient mucus-penetrating ability. Given the observed ability of Pep/Gal-PNPs to penetrate the mucus barrier, the next obstacle that we interrogated was cellular uptake.

## Pep/Gal-PNPs uptake by Caco-2 cells

To begin our cellular uptake analysis of Pep/Gal-PNPs, we first evaluated the cytotoxicity of these nanoparticles on human colorectal adenocarcinoma cells (Caco-2) and found no negative impact at concentrations of 0.05–1 mg mL$^{-1}$ (Supplementary Fig. 10). Therefore, subsequent studies were conducted in that concentration range. The cell-penetrating segment of Pep (CPP, R$_6$) was demonstrated to be pH-insensitive (Supplementary Fig. 11). Thus, the non-switchable nanoparticles (CPP/Gal-PNPs) modified with CPP and Gal were developed as a comparator (Supplementary Table 1), aiming to verify the cell-penetrating ability of Pep ligand and ascertain the advantages of ligand-switchable features of Pep/Gal-PNPs. Moreover, CPP/Gal-PNPs showed potent, pH-insensitive hemolytic activity (Supplementary Fig. 11). Studies on cellular uptake of nanoparticles by Caco-2 cells at pH 6.0 to 8.0 indicated that Pep/Gal-PNP internalization increased markedly as the pH decreased, while PNP and CPP/Gal-PNP internalization was consistent across the pH gradient (Supplementary Fig. 12). Given these results, we compared the cellular uptake efficiency of nanoparticles at pH 6.8 and 7.4. Confocal laser scanning microscopy (CLSM) images revealed stronger fluorescence signals for CPP/Gal-PNPs and Pep/Gal-PNPs than PNPs at pH 6.8, whereas Pep/Gal-PNP fluorescence decreased markedly at pH 7.4, in contrast with no change in the other groups (Fig. 4a). The quantitative analysis revealed that cellular uptake of insulin in Pep/Gal-PNPs was 2.6-fold higher at pH 6.8 than at pH 7.4 and almost 4.7-fold higher than that of PNPs (Fig. 4b). In contrast, free insulin could hardly be taken up by cells (Supplementary Fig. 13). These results indicated that Pep was exposed on the surface in response to mildly acidic pH and promoted the endocytosis of Pep/Gal-PNPs, leading to the intracellular delivery of insulin.

To better understand the effect of surface ligands on cellular uptake, the endocytosis mechanism of Pep/Gal-PNPs at different pH values was investigated. The uptake of Pep/Gal-PNPs was significantly decreased at 4 °C, suggesting that the endocytosis pathway was energy-dependent (Supplementary Fig. 14). To further interrogate the

mechanism, the endocytic inhibitors amiloride (macropinocytosis), chlorpromazine (clathrin-mediated endocytosis), and filipin (caveolae-mediated endocytosis) were used. Pep/Gal-PNP internalization by Caco-2 cells markedly decreased (to 49.6%) upon pretreatment with amiloride compared to control at pH 7.4, whereas no significant difference was observed at pH 6.8 (Supplementary Fig. 14). Compared to the control, chlorpromazine significantly decreased the uptake of Pep/Gal-PNPs to 39.7% at pH 6.8 and 61.6% at pH 7.4, respectively. By contrast, filipin had negligible effects on Pep/Gal-PNP uptake (Supplementary Fig. 14). These results revealed that ligand-switchable Pep/Gal-PNPs mainly adopted a clathrin-dependent endocytosis pathway at pH 6.8, whereas macropinocytosis predominated once Pep folded and got inactivated at physiological pH.

## Intracellular trafficking and transcytosis of Pep/Gal-PNPs

After entering cells, nanoparticles are typically transferred from endosomes to lysosomes for degradation[26]. Thus, we used CLSM to investigate the colocalization of nanoparticles with lysosomes. Interestingly, all the PNP, CPP/Gal-PNP, and Pep/Gal-PNP groups showed weak colocalization signals with lysosomes after incubation for 2 h (Fig. 4c), suggesting the capture of few nanoparticles. The hemolytic assay demonstrated the membrane-disrupting capabilities of CPP/Gal-PNPs and Pep/Gal-PNPs. Moreover, histidine protonation in Pep could also promote the release of Pep/Gal-PNPs from the lysosome[27]. Overall, these results indicated that Pep/Gal-PNPs could escape from lysosomes during intracellular trafficking, thereby protecting encapsulated insulin from degradation.

As Pep/Gal-PNPs appeared to remain intact intracellularly, the transcytosis efficiency of insulin by different formulations was next investigated. Pep/Gal-PNPs exhibited the highest apparent permeability coefficient (P$_{app}$) (9.62 ± 1.34 × 10$^{-6}$ cm s$^{-1}$) at pH 6.8, representing a 2.9-fold increase over that at pH 7.4 (Fig. 4d). In contrast, the P$_{app}$ values of PNPs and CPP/Gal-PNPs were not significantly different at pH 6.8 and 7.4 (Fig. 4d), and free insulin showed little transport across Caco-2 cells (Supplementary Fig. 13). The lack of a significant reduction in the transepithelial electrical resistance (TEER) of cells during transcytosis further confirmed that the nanoparticles underwent transcellular transport without opening the tight junctions of cells (Supplementary Fig. 15). Together, the results suggested that ligand-switchable Pep/Gal-PNPs promoted transcytosis of encapsulated insulin under mildly acidic conditions.

We next ascertained whether Pep/Gal-PNPs maintained structural integrity after exiting cells. FITC and RITC were encapsulated simultaneously in Pep/Gal-PNPs (FITC/RITC@NP) that were then incubated with cells for 2 h. An intense FRET spectrum was detected in the basolateral medium, suggesting that Pep/Gal-PNPs remained intact after exocytosis; this result was further confirmed by the cryo-TEM image (Fig. 4e). Then, we determined if the Pep still reserved pH sensitivity. The Edans- and Dabcyl-labeled Pep was used to prepare Pep/Gal-PNPs (Edans-Pep-Dabcyl-NP) which were incubated with cells and then collected from the basolateral medium. The emission intensity of these recovered Edans-Pep-Dabcyl-NP decreased markedly after the pH increased from 6.8 to 7.4 (Fig. 4f), yielding similar results as the FRET assay of Pep. These results demonstrated that Pep/Gal-PNPs remained unchanged during transepithelial transport, increasing the possibility of subsequent site-specific targeting.

## Selectivity of Pep/Gal-PNPs for hepatocytes

After verifying that Pep could facilitate the uptake of Pep/Gal-PNPs by intestinal epithelium, we further explored the functions of Gal ligands. Asialoglycoprotein receptors (ASGPRs) expressed on hepatocytes can specifically recognize Gal residues[28]. Therefore, the human fetal hepatocytes (LO2 cells) that expressed high levels of ASGPRs (Supplementary Fig. 16) were used as the cell model to study the interaction of Pep/Gal-PNPs with hepatocytes. Moreover, the Pep/Gal-PNPs

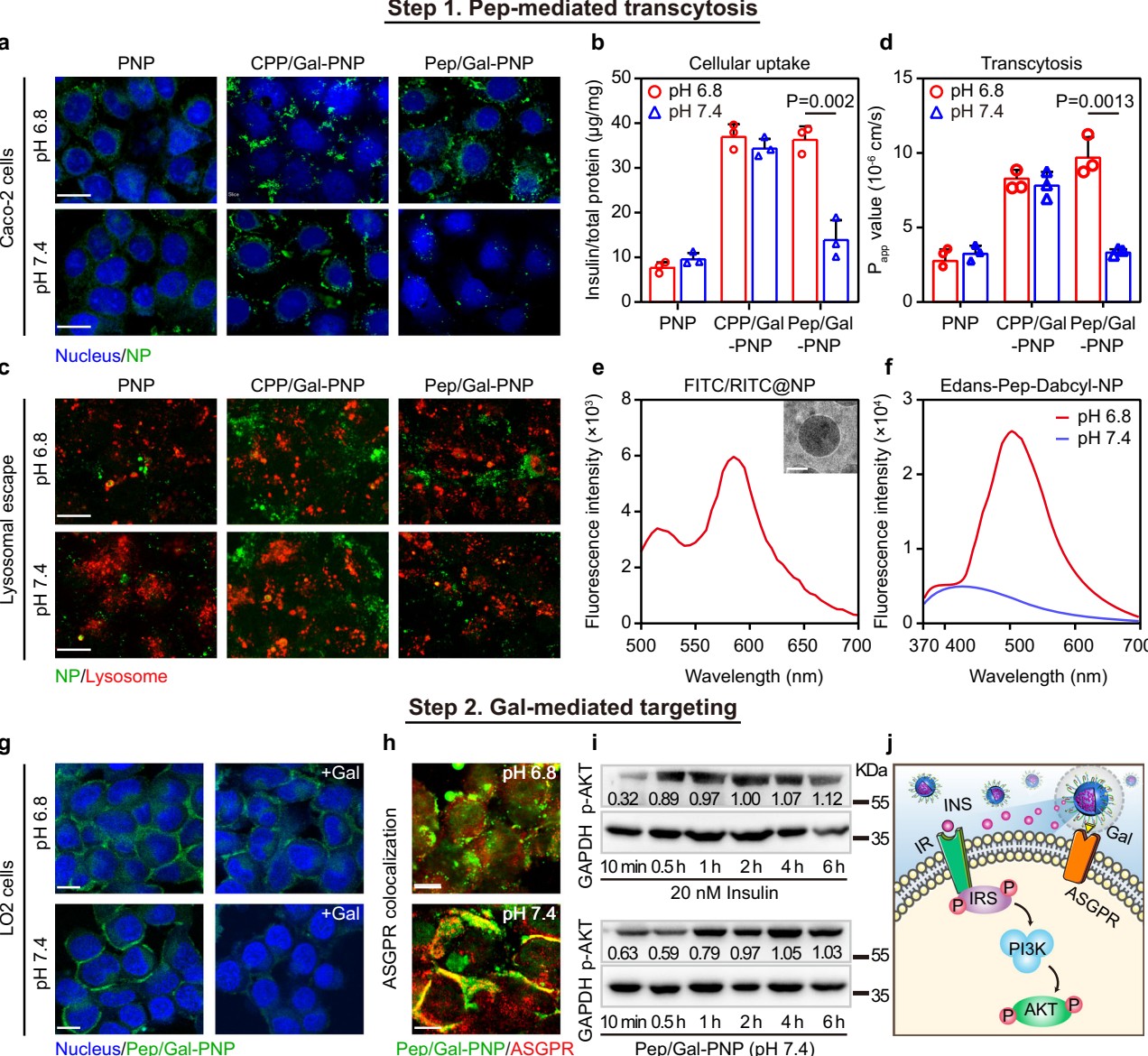

**Fig. 4 | In vitro transepithelial transport and hepatocyte selectivity of Pep/Gal-PNPs. a** Cellular uptake of nanoparticles by Caco-2 cells. Scale bar: 10 μm. **b** Quantitative analysis of nanoparticles internalized by Caco-2 cells. Data are presented as the mean ± SD ($n$ = 3 biologically independent experiments). **$p$ = 0.002, two-tailed Student's $t$-test. **c** Confocal laser scanning microscopy (CLSM) images of the colocalization of nanoparticles with lysosomes. Scale bar: 10 μm. **d** Apparent permeability coefficient ($P_{app}$) values for nanoparticle transport across Caco-2 cells. Data are presented as the mean ± SD ($n$ = 3 biologically independent experiments). **$p$ = 0.0013, two-tailed Student's $t$-test. **e** Emission spectrum and cryo-TEM image of FITC/RITC-loaded Pep/Gal-PNPs (FITC/RITC@NP) collected from the basolateral medium. Scale bar: 100 nm.Scale bar: 100 nm. **f** Emission spectra of Edans/Dabcyl-

labeled Pep-modified Pep/Gal-PNPs (Edans-Pep-Dabcyl-NP) collected from the basolateral medium at different pH values. **g** CLSM images of Pep/Gal-PNPs binding to LO2 cells. +Gal: in the presence of free galactose. Scale bar: 10 μm. **h** The colocalization of Pep/Gal-PNPs with ASGPRs on LO2 cells at different pH values. Scale bar: 10 μm. **i** Western blot analysis of p-AKT levels in LO2 cells after incubation with free insulin or insulin-loaded Pep/Gal-PNPs at pH 7.4 for the indicated time. The numbers represent the quantitative results of p-AKT levels normalized to GAPDH levels. The samples derived from the same experiment and blots were processed in parallel. **j** Schematic illustration of signaling in LO2 cells after exposure to Pep/Gal-PNPs at physiological pH. Source data are provided as a Source Data file.

exhibited negligible toxicity on LO2 cells (Supplementary Fig. 17). CLSM results indicated that Pep/Gal-PNPs exhibited strong fluorescence on LO2 cells regardless of pH (Fig. 4g). However, upon pre-incubation of the cells with free Gal, Pep/Gal-PNP fluorescence decreased markedly at pH 7.4 but not at pH 6.8 (Fig. 4g), indicating the important role of Gal in binding to LO2 cells. In contrast, the addition of Gal had no impact on PNP or CPP/Gal-PNP fluorescence at pH 6.8 or 7.4 (Supplementary Fig. 18). It might be ascribed to the non-selective cell-penetrating ability of CPP, which would enable the deposition of CPP/Gal-PNPs on hepatocytes regardless of pH. Then, we investigated the colocalization of Pep/Gal-PNPs with ASGPRs on LO2 cells and

found greater colocalization at pH 7.4 than at pH 6.8 (Fig. 4h). Accordingly, these results demonstrated that physiological pH triggered the ligand switching on the Pep/Gal-PNP surface to present Gal that specifically bound with ASGPRs on hepatocytes. As the hepatocyte is the main site for endogenous insulin to take effect[29], Pep/Gal-PNPs could precisely deliver encapsulated insulin to the aimed sites.

**Intracellular signaling upon Pep/Gal-PNPs binding**
The cellular effects of insulin are initiated at the cell membrane by binding to the insulin receptor (IR), which can stimulate the intracellular PI3K/AKT pathway and elicit the phosphorylation of AKT

(p-AKT)[30]. Thus, as an increase in intracellular p-ATK levels indicates the activation of IR-related signaling pathways, we investigated p-AKT levels in LO2 cells to evaluate the signaling potential of insulin delivered by Pep/Gal-PNPs. In this experiment, LO2 cells exposed to insulin-loaded Pep/Gal-PNPs at pH 7.4 showed a gradual increase over time in intracellular p-AKT levels, reaching the same level as the free insulin group after 2 h of treatment (Fig. 4i). We deduced that insulin was continuously released from Pep/Gal-PNPs and activated downstream intracellular pathways by binding to IR.

To further analyze the intracellular effects of insulin-loaded Pep/Gal-PNPs, modified pulse-chase p-AKT assays[31] were conducted to determine the sustained effect of these nanoparticles on LO2 cells, as previously reported. In this experiment, LO2 cells were first pulsed with free insulin and insulin-loaded Pep/Gal-PNPs for 30 min, then washed and chased by incubation in insulin-free DMEM. The results revealed that 4 h after the Pep/Gal-PNPs were removed, the cells still expressed high p-AKT levels, whereas p-AKT could not be detected in free insulin group after discarding the insulin solution (Supplementary Fig. 19). These results demonstrated that intracellular AKT phosphorylation is induced mainly by insulin released from cell-bound Pep/Gal-PNPs due to the Gal–ASGPRs interaction. In summary, the ligand-switchable Pep/Gal-PNPs bound to LO2 cells via interactions between the exposed Gal and ASGPRs at physiological pH, whereby they served as an insulin reservoir for the sustained activation of intracellular IR-related signaling pathway (Fig. 4j).

### In vivo intestinal absorption of Pep/Gal-PNPs

To investigate intestinal absorption of Pep/Gal-PNPs, the real-time transport of nanoparticles into intestinal villi of rats was studied using two-photon microscopy (TPM), which offers higher imaging depth with less photodamage[32]. After treatment for 30 min, much stronger Pep/Gal-PNP fluorescence was observed in intestinal villi at pH 6.8 than at pH 7.4. By contrast, similar intense CPP/Gal-PNP fluorescence was observed at pH 6.8 and 7.4 (Fig. 5a). However, non-functionalized PNPs consistently exhibited weak fluorescence signals in intestinal villi (Supplementary Fig. 20). CLSM images of intestinal sections further confirmed the greater intestinal absorption of Pep/Gal-PNPs at pH 6.8. FITC and RITC were simultaneously encapsulated in Pep/Gal-PNPs, and the colocalization of these two signals revealed the structural integrity of these nanoparticles (Fig. 5a). Quantitative analysis revealed that the relative integrated density of Pep/Gal-PNPs was 3.6-fold higher at pH 6.8 than at pH 7.4, but no significant difference was detected for CPP/Gal-PNPs (Fig. 5b). These results confirmed the better intestinal absorption of intact Pep/Gal-PNPs at the simulated intestinal pH, which was conducive to further delivering encapsulated drugs to the specific target sites.

### In vivo liver accumulation and selectivity of Pep/Gal-PNPs

After crossing intestinal barriers, Pep/Gal-PNPs could enter systemic circulation via portal vein at physiological pH. Therefore, we further investigated the tissue distribution of Pep/Gal-PNPs in vivo. At 4 h after oral administration of FITC-labeled nanoparticles to rats, the Pep/Gal-PNP fluorescence intensity in liver homogenates was much higher than in other organs, which was 1.66-fold higher than that of CPP/Gal-PNP group (Fig. 5c). The amount of Pep/Gal-PNPs accumulated in the liver and kidney was almost 71.9% ± 3.7% and 12.5% ± 1.5% of the total absorption. In contrast, the ratio of CPP/Gal-PNPs that accumulated in the kidney increased to 23.0% ± 4.0% (Supplementary Fig. 21). Additionally, we imaged the organs of rats using an in vivo imaging system (IVIS) to observe the biodistribution of nanoparticles directly. Pep/Gal-PNPs showed relatively more intense fluorescence in the liver compared with other organs, whereas CPP/Gal-PNPs exhibited strong fluorescence in the liver, lungs, and spleen (Fig. 5d). In contrast, PNPs showed weak fluorescence in all organs except the intestine (Supplementary Fig. 22), indicating limited nanoparticles transported across intestinal barriers. Overall, these results suggested that the Pep/Gal-PNPs mainly accumulated in the liver.

To further confirm that Pep/Gal-PNPs can specifically target the liver, the colocalization of nanoparticles with ASGPRs on hepatocytes in liver sections was detected using CLSM after immunofluorescence staining. Greater colocalization with ASGPRs was observed for Pep/Gal-PNPs than for CPP/Gal-PNPs (Fig. 5e). The colocalization coefficient (R) was calculated to be 0.03 for CPP/Gal-PNP group and 0.32 for Pep/Gal-PNP group, representing a 10.7-fold increase. In contrast, PNPs exhibited weak fluorescence in liver sections with little colocalization signals (Supplementary Fig. 23). The limited colocalization of CPP/Gal-PNPs with ASGPRs was mainly attributed to the non-selectivity of CPP, which compromised the targeting efficiency of Gal and allowed the nanoparticles to be captured by other liver cells (e.g., endothelial and Kupffer cells) and organs. In contrast, on the surface of Pep/Gal-PNPs, Gal was deshielded at physiological pH since Pep folded, enabling specific Gal binding to ASGPRs on hepatocytes.

### In vivo ligand-switching features of Pep/Gal-PNPs

To clarify the sequential intestinal barrier-crossing and liver-targeting abilities of Pep/Gal-PNPs, we further studied in vivo surface ligand-switching on Pep/Gal-PNPs by applying the FRET technique. A FRET pair, the carboxyfluorescein (FAM) as the donor and the carboxyte-tramethylrhodamine (TAMRA) as the acceptor, was conjugated to amino acid side groups of N- and C-termini of Pep, respectively. The FRET pair-labeled Pep was further applied to prepare the Pep/Gal-PNPs (FR-Pep/Gal-PNPs). Rat intestine and liver segments were isolated at 2 h and 4 h, respectively, after the oral administration of FR-Pep/Gal-PNPs, and the FRET efficiency of the nanoparticles was detected. Although FR-Pep/Gal-PNPs exhibited strong fluorescence in the intestine, the FRET efficiency remained relatively low (approximately 11.6%) (Fig. 5f), highlighting the stretched structure of the Pep. By contrast, the FRET efficiency of FR-Pep/Gal-PNPs was markedly increased in the liver, with a nearly 5-fold increase in efficiency compared with the intestine. Consistent with the in vitro results, the in vivo FRET findings further confirmed that Pep underwent structural changes in response to environmental pH along the oral route from the intestine to the liver, enabling the switching of functional surface ligands on Pep/Gal-PNPs.

### Visualization of the systemic delivery route of Pep/Gal-PNPs in vivo

To thoroughly examine in vivo delivery route of Pep/Gal-PNPs after oral administration, FITC-labeled Pep/Gal-PNPs were visualized in the intestine and liver of a living rat using the confocal laser endomicroscopy (CLE). Widespread green fluorescence of Pep/Gal-PNPs was detected in small intestine villi 2 h after administration (Fig. 6a, top row), indicating efficient intestinal absorption of the nanoparticles. After 4 h, marked Pep/Gal-PNP accumulation was observed in the liver, as indicated by the intense fluorescence (Fig. 6a, bottom row). Moreover, Pep/Gal-PNP fluorescence was observed to gradually increase from blood vessels to hepatocytes in the deep scan images (Fig. 6a, denoted by arrows). We speculated that fenestrations in liver sinusoidal endothelial cells[33] enabled Pep/Gal-PNPs to traverse hepatic vessels and reach hepatocytes. Collectively, these results confirmed that ligand-switchable Pep/Gal-PNPs could sequentially transit across intestinal barriers and accumulate in the liver to deliver insulin to hepatocytes.

### In vivo hypoglycemic efficacy

In the culmination of seeing that the Pep/Gal-PNPs arrive at the liver and deliver insulin to hepatocytes, we further investigated whether these nanoparticles would lead to an effective physiological response. As an indicator of the pharmacodynamic (PD) profile of insulin-related formulations, the hypoglycemic effect was evaluated in type I diabetic rats based on the blood glucose level (BGL) after dosing. The BGL of

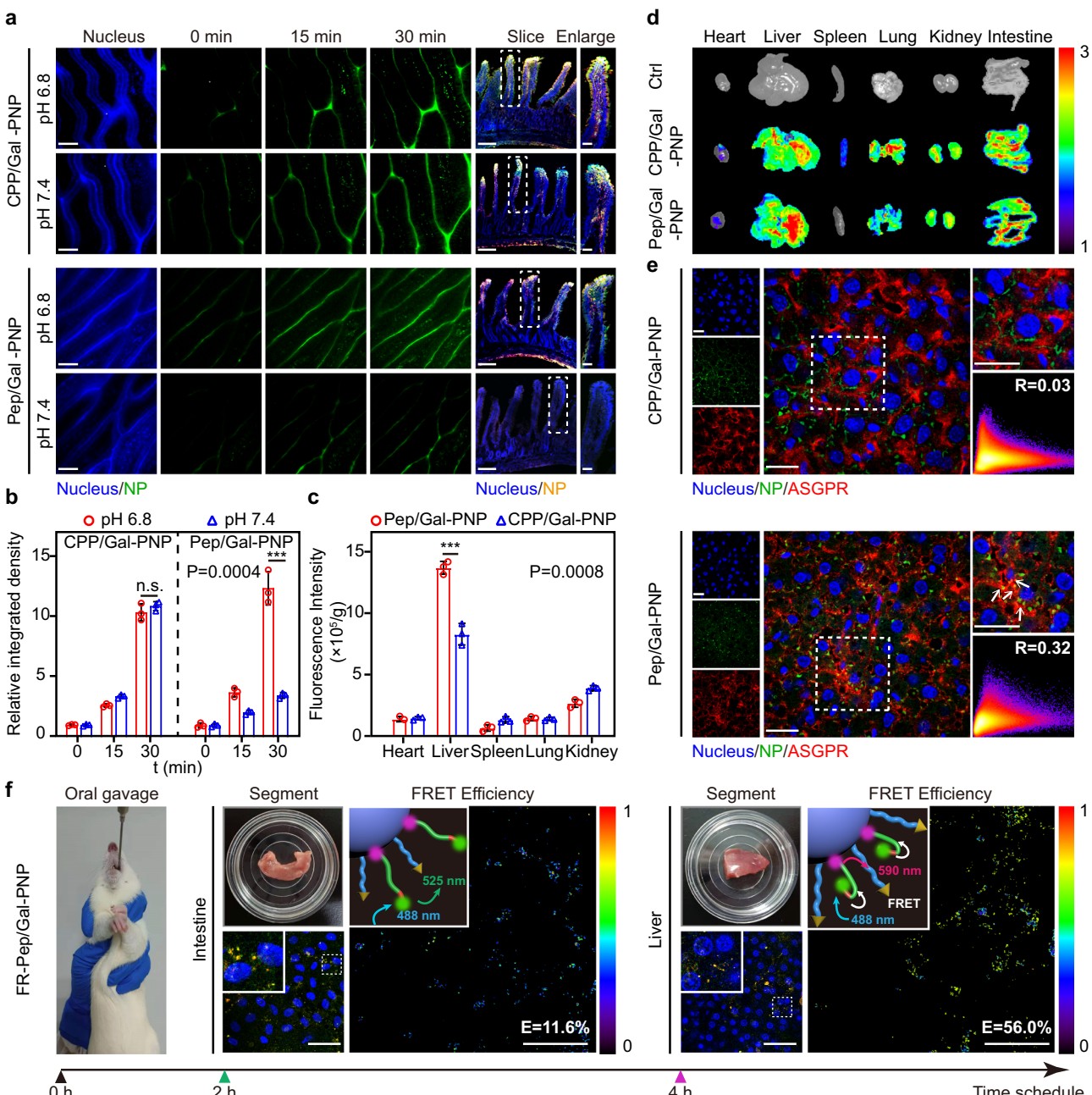

**Fig. 5 | In vivo sequential intestinal absorption and liver accumulation of Pep/Gal-PNPs. a** Two-photon microscopy (TPM) images show the absorption of nanoparticles in intestinal villi. Scale bar: 100 μm. Confocal laser scanning microscopy (CLSM) images of intestinal villus sections. Scale bars: intestinal slice images, 200 μm; enlarged images, 50 μm. **b** Quantitative analysis of the absorption of nanoparticles in intestinal villi. Data are presented as the mean ± SD (n = 3 biologically independent experiments). n.s., not significant, \*\*\*p = 0.0004, two-tailed Student's t-test. **c** The fluorescence intensity of different tissue homogenates prepared from rats 4 h after the oral administration of FITC-labeled nanoparticles. Data are presented as the mean ± SD (n = 3 biologically independent experiments). \*\*\*p = 0.0008, two-tailed Student's t-test. **d** The accumulation of nanoparticles in major rat organs as imaged by in vivo imaging system (IVIS). Ctrl: rats treated with PBS. The color bar indicates the radiant efficiency × $10^7$ p s$^{-1}$ cm$^{-2}$ sr$^{-1}$. **e** The colocalization of nanoparticles with ASGPRs in liver sections. R: colocalization coefficient. Scale bar: 20 μm. **f** The distribution and FRET efficiency of FAM/TAMRA-labeled Pep-modified Pep/Gal-PNPs (FR-Pep/Gal-PNPs) in intestine and liver segments prepared from rats 2 h and 4 h after oral administration, respectively. Scale bar: 50 μm. Source data are provided as a Source Data file.

diabetic rats treated subcutaneously with free insulin dropped sharply to approximately 15.7% of the initial level at 3 h post-administration and then gradually returned to the basal level (Fig. 6b). Notably, wild fluctuations in blood glucose can cause severe side effects[34]. In contrast, insulin-loaded nanoparticles yielded more moderate and prolonged hypoglycemic effects (Fig. 6b). Pep/Gal-PNPs generated the most pronounced hypoglycemic effect among the three nanoparticle formulations, reaching a minimum BGL of 23.2% of the initial level at 8 h post-administration (Fig. 6b). Moreover, the BGL of rats treated

with Pep/Gal-PNPs remained within the normal range for 7 h (Supplementary Fig. 24). Correspondingly, Pep/Gal-PNPs were calculated to achieve the highest pharmacological availability (PA) of 10.1% (Supplementary Table 2).

Subsequently, the pharmacokinetic (PK) profiles of different formulations were investigated based on the serum insulin concentration over time. Consistent with the PD results, diabetic rats treated with subcutaneous insulin showed a sharp increase in peripheral serum insulin, which reached the maximum 1 h post-injection and rapidly

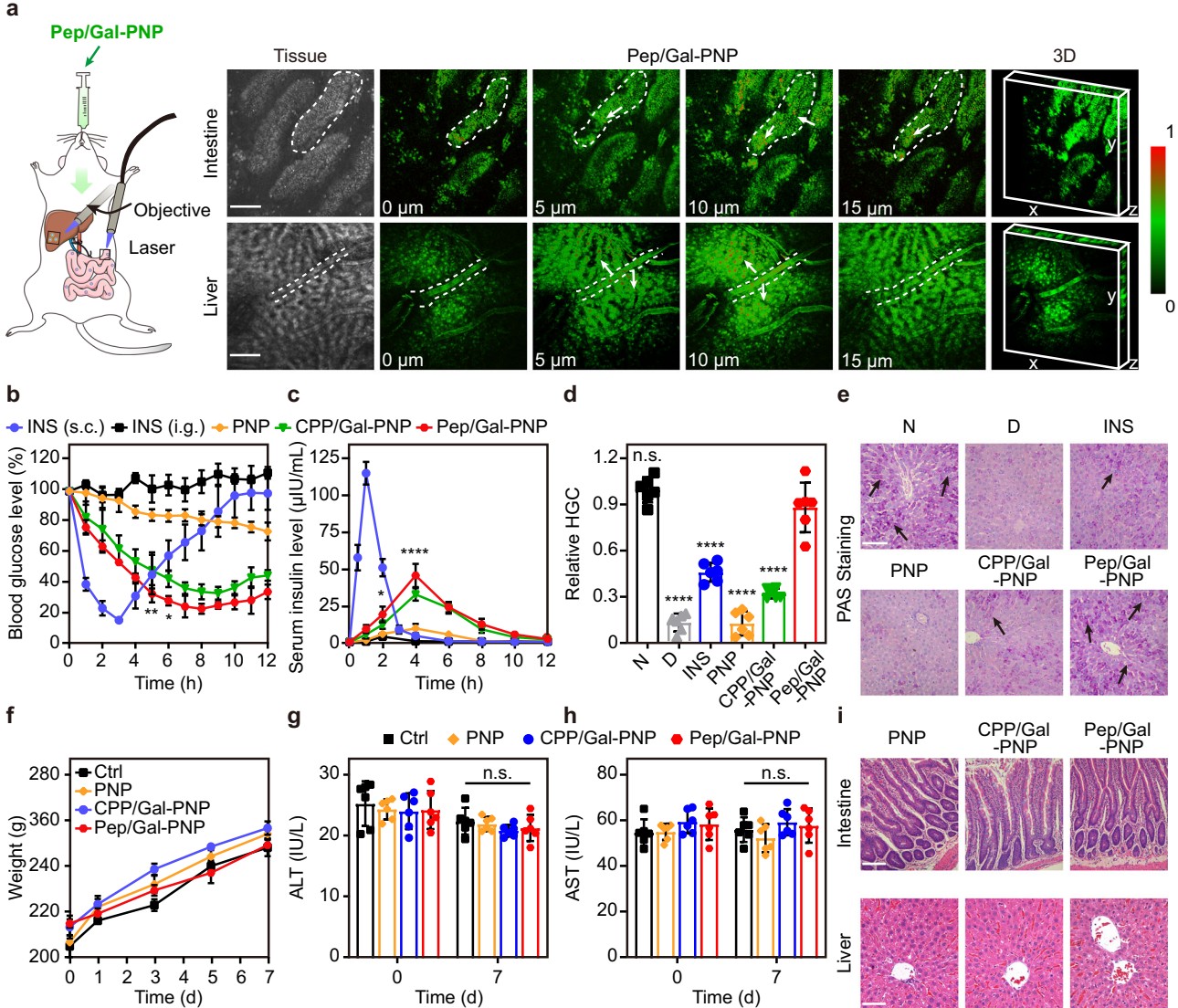

**Fig. 6 | In vivo trafficking, hypoglycemic effects, and toxicity of nanoparticles.**
**a** Confocal laser endomicroscopy (CLE) images of intestine villi (top row) and liver lobe (bottom row) from a rat after oral administration of FITC-labeled Pep/Gal-PNPs. Scale bar: 100 μm. The color bar indicates the fluorescence intensity (a.u.).
**b** Blood glucose levels over time in type I diabetic rats treated with different formulations. Data are presented as the mean ± SD ($n = 6$ biologically independent rats). **$p = 0.0054$ at 5 h, *$p = 0.0129$ at 6 h compared with the CPP/Gal-PNP group, two-way analysis of variance (ANOVA) with Tukey's post-hoc test. **c** Peripheral serum insulin levels over time in diabetic rats treated with different formulations. Data are presented as the mean ± SD ($n = 6$ biologically independent rats). *$p = 0.0316$ at 2 h, ****$p < 0.0001$ at 4 h compared with the CPP/Gal-PNP group, two-way analysis of variance (ANOVA) with Tukey's post-hoc test. **d** Relative hepatic glycogen content (HGC) in healthy rats treated with PBS (N); diabetic rats treated with PBS (D), insulin (INS), and insulin-loaded nanoparticle formulations (PNP, CPP/

Gal-PNP, and Pep/Gal-PNP). Data are presented as the mean ± SD ($n = 6$ biologically independent rats). n.s., not significant, ****$p < 0.0001$ compared with the Pep/Gal-PNP group, one-way analysis of variance (ANOVA) with Tukey's post-hoc test.
**e** Images of periodic acid-Schiff (PAS) staining of liver sections. The black arrows denote synthesized glycogen. Scale bar: 100 μm. **f** Average body weight of healthy rats treated with PBS (Ctrl) and nanoparticle formulations every day for a week. Data are presented as the mean ± SD ($n = 6$ biologically independent rats).
**g, h** Serum ALT and AST levels in rats treated with different formulations. Data are presented as the mean ± SD ($n = 6$ biologically independent rats). n.s., not significant compared with the Ctrl group, two-tailed Student's $t$-test. **i** Images of hematoxylin and eosin (H&E) staining of intestine and liver sections from rats treated with different formulations. Scale bar: 100 μm. Source data are provided as a Source Data file.

decreased to baseline in the following 3 h (Fig. 6c). Compared with the other nanoparticle formulations, Pep/Gal-PNPs achieved a considerably higher concentration of insulin at 4 h post-administration (Fig. 6c) and reached the highest relative oral bioavailability of insulin at 7.7% (Table 1). Notably, this result is comparable to some PLGA-based oral insulin nanoparticles[35].

**In vivo hepatic glucose utilization studies**
It has been reported that direct insulin delivery to the liver could promote hepatic glucose utilization and glycogen production in

diabetes[17]. Therefore, we evaluated hepatic glycogen storage in diabetic rats treated with different formulations. Quantitative analysis results indicated that orally administered insulin-loaded Pep/Gal-PNPs induced the highest level of liver glycogen synthesis, with a relative hepatic glycogen content (HGC) approximately 7.2-, 1.9- and 2.7-fold higher than that in the diabetic rats (D group), subcutaneous injection of insulin (INS group) and CPP/Gal-PNP group (Fig. 6d). Surprisingly, the Pep/Gal-PNP group had a hepatic glycogen level similar to that of the normal rats (N group) (Fig. 6d). Moreover, the glycogen synthesis in rats was directly observed by using periodic acid-Schiff (PAS)

**Table 1 | Pharmacokinetic parameters of different insulin formulations following oral or subcutaneous administration to diabetic rats**

|  | Insulin (s.c.) | Insulin (i.g.) | PNP (i.g.) | CPP/Gal-PNP (i.g.) | Pep/Gal-PNP (i.g.) |
|---|---|---|---|---|---|
| Dose (IU kg$^{-1}$) | 5 | 75 | 75 | 75 | 75 |
| AUC (μIU*h mL$^{-1}$)[a] | 182.2 ± 7.1 | 11.0 ± 1.0 | 41.6 ± 6.0 | 159.4 ± 10.4 | 210.6 ± 14.9 |
| $T_{max}$ (h)[b] | 1 | 4 | 4 | 4 | 4 |
| $F$ (%)[c] | 100 | 0.4 | 1.5 | 5.8 | 7.7 |

[a]AUC: area under the peripheral serum insulin level versus time curve; [b]$T_{max}$: time at which the maximum plasma insulin level was reached; [c] $F$: relative bioavailability. s.c., subcutaneous; i.g., intragastric (oral).
Data are presented as the mean ± SD ($n$ = 6).

staining. The D group presented depleted glycogen levels in the liver compared to the N group, and this depletion was hardly improved by treatment with PNPs (Fig. 6e). In contrast, large amounts of hepatic glycogen were detected in the Pep/Gal-PNP group compared with INS and CPP/Gal-PNP groups (Fig. 6e).

Since the insulin-loaded Pep/Gal-PNPs largely accumulated in the liver, we further demonstrated that these nanoparticles elicited the highest portal serum insulin levels among the formulations (Supplementary Fig. 25). Notably, the area under the curve (AUC) of the portal serum insulin level in Pep/Gal-PNP group was 4.1-fold higher than in the subcutaneous insulin group (Supplementary Table 3). Moreover, through simultaneously analyzing the peripheral and portal serum insulin levels in the same rats, the results indicated Pep/Gal-PNPs produced a ~4.2-fold increase in insulin level exposed to the liver than periphery (Supplementary Fig. 26), mimicking the biodistribution of endogenous insulin[36]. In contrast, this insulin gradient was lost in rats following subcutaneous insulin administration, which exhibited higher insulin concentration in the peripheral circulation and might cause hypoglycemia events. Additionally, we further calculated the hepatic insulin availability ($F_h$) of Pep/Gal-PNPs, and the results indicated it was significantly higher than their peripheral insulin bioavailability (19.9% vs. ~7%) (Supplementary Fig. 26). These results demonstrated that the Pep/Gal-PNPs could promote hepatic glycogen production in diabetic rats through elevating intrahepatic insulin exposure. In summary, we developed the reasonable hypothesis that ligand-switchable Pep/Gal-PNPs underwent efficient intestinal absorption to enable the subsequent hepatic deposition of insulin, which replicated the endogenous insulin pathway to reestablish a high portal–periphery insulin gradient, thereby promoting the conversion of blood glucose into glycogen for storage and maintaining glucose homeostasis.

### In vivo toxicity analysis

Finally, the in vivo toxicity of the Pep/Gal-PNPs was assessed by monitoring the body weight of rats after the oral administration of nanoparticles every day for a week. In these studies, the administration dose of nanoparticles (1000 mg kg$^{-1}$) was about 20-fold higher than their effective dose (50 mg kg$^{-1}$). The results showed no significant differences in the body weight of rats in the experimental groups compared to the control group (Fig. 6f). As an indicator of potential liver toxicity, serum alanine aminotransferase (ALT) and aspartate aminotransferase (AST) levels were detected in the experimental and control groups. Importantly, serum ALT and AST levels were within the normal range (ALT: 10 to 40 IU L$^{-1}$; AST: 50 to 150 IU L$^{-1}$)[37] for rats in all groups (Fig. 6g, h). Moreover, hematoxylin and eosin (H&E) staining revealed no histological damage in the intestine or liver of experimental rats compared with control rats (Fig. 6i, Supplementary Fig. 27). These results demonstrated that Pep/Gal-PNPs were biocompatible in vivo and thus suitable for oral insulin delivery.

### Discussion

Drug delivery systems with multiple functions are required to traverse complex physiological environments and target specific sites. Despite dual-ligand modified vehicles having been widely applied to meet these demands, mutual interference between surface ligands might be an important factor limiting their functionalities and thus resulting in low in vivo drug delivery efficiency[5,9]. Here, we rationally designed multifunctional nanoparticles (Pep/Gal-PNPs) with simultaneous modification of dual ligands. The Pep ligands on Pep/Gal-PNPs underwent conformational changes which extend from the surface at simulated intestinal pH (6.8) and folding at physiological pH (7.4), as demonstrated in in vitro and in vivo studies. Moreover, the AFM results indicated that the other Gal ligands on Pep/Gal-PNPs could be exposed on surface as the Pep folded at physiological pH. Therefore, the Pep/Gal-PNPs could switch dual-functional ligands to avoid their interference by adjusting conformations of Pep in response to environmental pH. Likewise, viruses coordinate diverse surface proteins by responsively modulating their conformations, thereby enabling sequential functions that support efficient invasion. In that way, the Pep/Gal-PNPs share similarities with viral unique surfaces. Compared with the reported virus-mimicking nanocarriers with virion components which might cause concerns of immunogenicity issues[38], the Pep/Gal-PNPs appear as a safer option. Although the nanoparticles could not emulate major functions of viruses (i.e., replication and tropism), the changes in surface functionalities and the subsequent full display of diverse functions make it similar to some enveloped viruses.

The ligand-switchable Pep/Gal-PNPs demonstrated the potential to sequentially exert functions of diverse surface ligands triggered by pH variations along the delivery route from the intestine to the liver after oral administration. Meanwhile, the Pep/Gal-PNPs could maintain stability in the harsh physiological environment due to the protection of the PEG layer. First, the Pep adopted a stretched conformation and got activated at simulated intestinal pH, which increased its exposure on the surface and promoted the intestinal absorption of Pep/Gal-PNPs. The results of cellular experiments and two-photon microscopy examination demonstrated that Pep/Gal-PNPs could efficiently overcome intestinal barriers which typically functions as the first line of defense to restrict nanoparticles from entering blood circulation[39,40]. Then, after traversing the intestinal barriers, most of the nanoparticles are delivered to the portal vein, which harbors a physiological pH. Therefore, Gal was deshielded on the surface of Pep/Gal-PNPs as Pep folded in this microenvironment. Although most nanoparticles inevitably reach the liver after oral administration, Pep/Gal-PNPs could selectively target hepatocytes through the binding of Gal to ASGPRs on cells which is comparable to that of previously reported Gal-modified nanoparticles[41]. In contrast, although CPP/Gal-PNPs modified with the cell-penetrating segment of Pep exhibited similar enhanced intestinal absorption efficiency, they lacked cell specificity in the liver and increased the accumulation ratios in other organs, which was attributed to their non-switchable surface ligands. Taken together, the ligand-switchable nature of Pep/Gal-PNPs could realize the full potential of the dual surface ligands, ultimately delivering encapsulated drugs to hepatocytes in the liver with high efficiency. Though the pH-responsive cell-penetrating peptides have been studied in previous publications[42], we thoroughly characterized their conformational changes and stretchable structures and applied them to develop multifunctional ligand-switchable nanocarriers, which have not been

reported previously. Moreover, it is noteworthy that this work presents a promising strategy to fully exert diverse functions of dual surface ligands on nanocarriers, which is distinct from reported ligand-switchable nanocarriers that hide the ligands until reaching the specific sites[43,44].

As a proof-of-concept, we utilized insulin as the model drug. In vivo therapeutic assessments revealed that the insulin-loaded Pep/Gal-PNPs elicited a sustained and strong hypoglycemic response in diabetic rats. Moreover, the insulin-loaded Pep/Gal-PNPs could increase insulin deposition in the liver and restore the liver–periphery insulin gradient in diabetes. It has been reported that the liver is exposed to approximately 2- to 4-fold higher concentrations of endogenous insulin than peripheral tissues (such as brain and fat) under normal circumstances[36]. However, this physiological distribution would be disrupted by conventional subcutaneous injection of insulin, which may lead to peripheral hyperinsulinemia and severe hypoglycemia[45]. Additionally, the hepatoselective insulin analogues could elicit overly hepatic insulinization and lead to abnormal hepatic lipid accumulation which might increase the risk of liver steatosis[29,46]. In comparison, the Pep/Gal-PNPs could mimic the biodistribution of endogenous insulin by actively targeting insulin to the liver, simultaneously maintaining optimal insulinization in the liver and the periphery. Therefore, Pep/Gal-PNPs show advantages for effective diabetes treatment and reduce the risk of adverse effects compared with hepatoselective insulin analogues[47].

Furthermore, we investigated the mechanism of action of the insulin-loaded Pep/Gal-PNPs on the liver. In this study, we showed that insulin-loaded Pep/Gal-PNPs could sustainably activate PI3K/AKT signaling pathway in hepatocytes that regulate glucose utilization and glycogen storage[48]. Therefore, these results reminded us that insulin-loaded Pep/Gal-PNPs had the potential to promote hepatic glycogen synthesis. Although oral insulin therapy is generally considered to have advantages in promoting hepatic glycogen synthesis[49], these findings explicate that intracellular mechanism. Correspondingly, our animal experiments further demonstrated that compared with the conventional subcutaneous injection of insulin, orally administered insulin-loaded Pep/Gal-PNPs showed greater efficacy in promoting the liver to take up and store glucose as glycogen due to superior liver selectivity. Surprisingly, the hepatic glycogen level in Pep/Gal-PNP-treated diabetic rats was similar to that in healthy rats, suggesting that this treatment could potentially correct defects in glucose metabolism in diabetes. Most recent studies of oral insulin therapy mainly focus on lowering blood glucose. However, dramatic fluctuations in BGLs are more deleterious than stable high glucose concentrations[49]. The liver glycogen plays a critical role in defending against hypoglycemia[50]. Whereas, hepatic glycogen storage is impaired in diabetes, which restricts the ability of hepatocytes to respond appropriately to glucose levels[17]. Therefore, Pep/Gal-PNPs have the potential to maintain glycemic homeostasis rather than merely lowering BGL in the context of diabetes. The outcomes of this study emphasize the importance of hepatic glycogen for diabetes management, and the ligand-switchable Pep/Gal-PNPs may represent a significantly improved oral insulin therapy.

In summary, we have rationally developed ligand-switchable nanoparticles (Pep/Gal-PNPs) that realize the full potential of diverse surface ligands in response to environmental pH, resembling the unique surface features of viruses. Pep/Gal-PNPs sequentially overcome intestinal barriers and target insulin to the liver in response to variations in pH after oral administration, thereby promoting hepatic glycogen production to maintain glucose homeostasis as improved oral insulin therapy. Moreover, this study provides a promising strategy for the effective functionalization of nanocarriers with diverse ligands, which exhibit tremendous potential for a broad range of drug-delivery applications in the future, such as biomacromolecules and antitumor drugs.

## Methods

### Experimental reagents
Poly (D, L-lactide-co-glycolide)-carboxylic acid (PLGA-COOH, LA/GA molar ratio 50:50, Mw ~15,000 Da) was purchased from Daigang Bio-material Co., Ltd. (Jinan, China) and the PLGA-maleimide (PLGA-Mal, LA/GA molar ratio 50:50, Mw ~15,000 Da) was synthesized by Ruixi Biological Technology Co., Ltd. (Xi'an, China). Diamino-poly (ethylene glycol) (NH$_2$-PEG-NH$_2$, Mw ~1,000 Da) was purchased from Ponsure Biological Technology Co., Ltd. (Shanghai, China). The pH-triggered stretchable cell-penetrating peptide (Pep) and FRET pair-labeled Pep was synthesized by BankPeptide Biological Technology Co., Ltd. (Hefei, China). Human insulin was the gift received from Novo Nordisk A/S. Fluorescein isothiocyanate (FITC), 3-(4, 5-Dimethyl-2-thiazolyl)-2, 5-diphenyl-2H-tetrazolium bromide (MTT), rhodamine isothiocyanate (RITC), 2-(4-amidinophenyl)-6-indolecarbamidine dihydrochloride (DAPI), radioimmunoprecipitation assay (RIPA), Lyso-Tracker red and bicinchoninic acid (BCA) Protein Assay Kit were all purchased from Meilun Biotechnology Co., Ltd. (Dalian, China). Hoechst 33258, Hoechst 33342, and Alexa 647 labeled goat anti-rabbit IgG were purchased from Yeasen Biotechnology Co., Ltd. (Shanghai, China). Anti-ASGPR rabbit polyclonal antibody (pAb), anti-GAPDH mouse pAb, horseradish peroxidase (HRP)-conjugated goat anti-rabbit IgG, and HRP-conjugated goat anti-mouse IgG were all purchased from Sangon Biotechnology Co., Ltd. (Shanghai, China). Phospho-AKT (Ser473) rabbit monoclonal antibody (mAb) was purchased from Bimake Co., Ltd. (Houston, USA). Human insulin ELISA kits were purchased from Mercodia (Uppsala, Sweden), glycogen ELISA kits were purchased from Solarbio Science and Technology Co., Ltd. (Beijing, China), and ALT and AST assay kits were purchased from Nanjing Jiancheng Bioengineering Co., Ltd. (Nanjing, China). All the other chemicals were of analytical grade and were purchased from Sinopharm Chemical Reagent Co., Ltd. (Shanghai, China).

### Cell culture
Caco-2 and LO2 cell lines were purchased from the American Type Culture Collection (Manassas, USA). HT29-MTX-E12 (E12) cell line was kindly provided by Novo Nordisk A/S (Denmark). Caco-2 and LO2 cells were maintained in Dulbecco's Modified Eagle medium (DMEM) with 5% (v/v) fetal bovine serum (FBS), 1% penicillin, and streptomycin (100 IU mL$^{-1}$) at 37 °C in 5% CO$_2$. E12 cells were maintained in DMEM with 10% (v/v) FBS, 1% (v/v) nonessential amino acids, 1% penicillin, and streptomycin (100 IU mL$^{-1}$) at 37 °C in 5% CO$_2$.

### Animal care
Male Sprague–Dawley (SD) rats (200–220 g) were provided by the Animal Experiment Center of Shanghai Institute of Materia Medica (Shanghai, China). All animal experiments were conducted following the Institutional Animal Care and Use Committee (IACUC) guidelines of the Shanghai Institute of Materia Medica (IACUC code: 2020-05-GY-58). To induce type I diabetes, the rats were fasted overnight before studies but allowed free access to water, and then injected intraperitoneally with 10 mM streptozotocin at a dose of 65 mg kg$^{-1}$. The rats with fasting blood glucose levels higher than 300 mg dL$^{-1}$ were regarded as diabetic.

### Characterization of pH-triggered stretchable Pep
First, the synthesized Pep was dissolved in a mixture of water/acetonitrile/acetic acid (87:8:5) and analyzed via electrospray ionization mass spectrometry (ESI-MS; QTRAP 4500, AB SCIEX, USA). Subsequently, 0.5 mg mL$^{-1}$ Pep was incubated in PBS at pH 3.0, 5.0, 6.0, 6.8, 7.0, 7.4, and 8.0. The secondary conformation of Pep under different pH conditions was measured using circular dichroism (CD; J-810, JASCO, Japan) and analyzed by Spectra Manager software (JASCO, Japan). To investigate the structural changes in Pep in response to pH, Pep was modified at the N- and C-termini with 5-[(2-aminoethyl)amino]

naphthalene-1-sulfonic acid) (Edans) and 4-(4-dimethylaminopheny-lazo)benzoic acid (Dabcyl) (a FRET pair), respectively. This FRET pair-labeled Pep was suspended in PBS at different pH values with the final concentration of $1\,mg\,mL^{-1}$, and the FRET emission of the Pep at 340 nm was measured by a microplate reader (Synergy H1, BioTek, USA).

### Synthesis of PLGA-Pep polymers
The Pep was covalently bound to the PLGA-Mal polymers based on the Michael-type addition reaction. A cysteine was introduced to the C-terminal of the Pep to offer a thiol group that could react with the maleimide group of the PLGA-Mal polymers. Concisely, the PLGA-Mal (750 mg, 0.05 mmol) and Pep (200 mg, 0.05 mmol) were dissolved in N, N-dimethylformamide (DMF, 4 mL) and stirred overnight at room temperature. Then the final solution was purified by dialysis (Mw cutoff: 10 KDa) against deionized water, and the final solution was lyophilized to obtain the PLGA-Pep polymers. The polymers were dissolved in hexadeuterodimethyl sulfoxide (DMSO-d6) and analyzed by $^1$H NMR spectroscopy (Avance III 500, Bruker, Switzerland).

### Synthesis of PLGA-PEG-Gal polymers
The PLGA-PEG-Gal polymers were synthesized by conjugating NH$_2$-PEG-Gal (Supplementary Method 4) with PLGA-COOH. The PLGA-COOH (423 mg, 0.028 mmol) was first dissolved in 2 mL DMSO, followed by adding 1-ethyl-3-(3-dimethylaminopropyl)-carbodiimide (EDC, 28 mg, 0.14 mmol) and N-hydroxysuccinimide (NHS, 16 mg, 0.14 mmol). After stirring for 15 min, the NH$_2$-PEG-Gal (34 mg, 0.028 mmol) was added and continued to react overnight. The resulting solution was purified by the method mentioned above. The obtained PLGA-PEG-Gal polymers were dissolved in DMSO-d6 and analyzed by $^1$H NMR spectroscopy.

### Preparation and characterization of nanoparticles
The nanoparticles, including PNPs, Pep-PNPs, Gal-PNPs, CPP/Gal-PNPs, and Pep/Gal-PNPs, were prepared using a modified double emulsion and solvent evaporation method[51]. In brief, the functional polymers were dissolved in 2 mL of dichloromethane (DCM) as the organic phase. Then, 0.2 mL of human insulin (dissolved in 0.01 M HCl) or aqueous sodium dodecyl sulfate (SDS, 0.05%, w/v) was emulsified with the organic phase by sonication (100 W, 30 s) to prepare the primary emulsion, which was subsequently added to 10 mL of 0.05% SDS and sonicated under the same conditions. The residual organic solvent was removed via vacuum evaporation. The nanoparticles were washed 3 times with PBS by centrifugation (9391 × g, 5 min) to remove unloaded insulin. The insulin was replaced with FITC-insulin to prepare FITC-labeled nanoparticles, or FITC and RITC in PBS (1 mg mL$^{-1}$, 50 μL) were added simultaneously to prepare FITC/RITC-labeled nanoparticles.

The size and zeta potential of nanoparticles suspended in PBS at pH 3.0, 5.0, 6.0, 6.8, 7.0, 7.4, and 8.0 were measured using a Zetasizer (Nano ZS, Malvern Instruments, UK). Nanoparticle morphology was observed by cryogenic transmission electron microscopy (cryo-TEM; TF20, FEI, USA) with an acceleration voltage of 200 kV. The nanoparticles were demulsified using acetonitrile, then the entrapment efficiency (EE) and loading capacity (LC) of insulin in nanoparticles were quantified by high-performance liquid chromatography (HPLC; Agilent 1260, USA) and calculated using the following equation:

$$Loading\ capacity\ (LC\%) = \frac{Total\ amount\ of\ insulin\ loaded - free\ insulin}{Total\ amount\ of\ insulin} \quad (1)$$

$$Entrapment\ efficiency\ (EE\%) = \frac{Total\ amount\ of\ insulin\ loaded - free\ insulin}{Nanoparticles\ weight} \quad (2)$$

The conjugation efficiency of Gal, Pep, and CPP to the surface of nanoparticles was measured using the resorcinol/sulfuric acid micro-method and BCA assay, as previously reported[28].

### Detection of nanoparticles by AFM
The nanoparticles were fixed on the silica substrate using our previously reported method[52]. Then, the substrate was immersed in buffer at pH 7.4 and scanned via AFM (FastScan Bio, Bruker, Germany) using a silicon probe (Bruker, Germany) at a rate of 1 Hz (256 samples per line) at 37 °C. Then, the probe was lifted, and the medium was discarded and replaced with buffer at pH 6.8 without moving the substrate. After incubation for 10 min, the substrate was scanned repeatedly by AFM under the same conditions. For the detection of ligand corona around nanoparticles, more than 10 nanoparticles were carefully examined for each group. Height-map images were handled for 3D reconstruction, and the height profiles were processed by NanoScope Analysis software (Bruker, Germany).

### Cellular uptake studies on Caco-2 Cells
The Caco-2 cells were seeded on 24-well plates and cultured for 2 days. The FITC-labeled nanoparticles were diluted with PBS (pH 6.8 or 7.4) to maintain the same dose of encapsulated insulin at 20 ug mL$^{-1}$. The cells were incubated with FITC-labeled nanoparticles for 2 h. Then the cells were washed with PBS and disrupted by RIPA lysis buffer. The amount of insulin in the lysate was detected using the microplate reader, and the total protein was quantified by the BCA kit. For CLSM observation, the Caco-2 cells were seeded on the microscope slides for 2 days. The cells were incubated with FITC-labeled nanoparticles for 2 h. Then the cells were washed, fixed with 4% paraformaldehyde, and stained with DAPI for 10 min. The cellular uptake of nanoparticles was observed using CLSM (FV1000, Olympus, Japan).

### Intracellular fate of nanoparticles
In brief, the Caco-2 cells were stained with Hoechst 33342 and Lyso-Tracker Red at 37 °C for 30 min. Then FITC-labeled nanoparticles at pH 6.8 and 7.4 were added to cells, which were incubated for another 2 h. Afterward, the colocalization signals of nanoparticles with lysosomes were imaged by CLSM.

### Transcellular transport studies
To investigate the transcytosis efficiency of nanoparticles, the Caco-2 cells were seeded on the 12-well transwell plates and continuously cultured for 21 days to mimic the intestinal epithelium monolayer. The cells were incubated with FITC-labeled nanoparticles at pH 6.8 and 7.4, respectively. Then, 0.2 mL of sample from the basolateral chamber was removed at predetermined time intervals (0.5, 1, 1.5, 2 h), and an equal amount of PBS was supplemented to maintain the volume. Meanwhile, TEER values of cells were measured using an electrical resistance meter (Millicell ERS-2, Millipore). The FITC-insulin was quantified using the microplate reader, and the $P_{app}$ values of insulin in different formulations were calculated using the following equation:

$$P_{app} = \frac{dQ}{dt} \times \frac{1}{A \times C_0} \quad (3)$$

where dQ/dt is the flux of insulin from the apical to the basolateral chamber, A is the diffusion membrane area (cm$^2$), and $C_0$ is the initial concentration of insulin in the donor compartment.

FRET assays were conducted to evaluate the integrity and pH sensitivity of the Pep/Gal-PNPs after exocytosis. The FITC/RITC-labeled Pep/Gal-PNPs (FITC/RITC@NP) were incubated with Caco-2 cells at pH 6.8 for 2 h. Subsequently, the basolateral sample was collected, and the FRET emission of the sample was detected at 450 nm by the microplate reader. In addition, the Pep/Gal-PNPs collected from the basolateral sample were also observed using cryo-TEM. The Edans/Dabcyl-labeled

Pep was utilized to prepare Pep/Gal-PNPs (Edans-Pep-Dabcyl-NP) as described above. After incubation with Caco-2 cells for 2 h, the basolateral sample was collected and adjusted to pH 6.8 or 7.4, and then the emission spectra of the sample were measured with an excitation wavelength at 340 nm by the microplate reader.

### Interaction of Pep/Gal-PNPs with hepatocytes

LO2 cells were seeded on microscope slides and cultured for 2 days. The cells were incubated with FITC-labeled nanoparticles at pH 6.8 and 7.4 for 2 h. To investigate the effect of Gal on the interaction of nanoparticles with cells, LO2 cells were pretreated with 50 μM Gal for 1 h at 37 °C before incubation with nanoparticles. Then, the cells were stained with DAPI and observed by CLSM.

To investigate the colocalization of Pep/Gal-PNPs with ASGPRs on LO2 cells, the cells were incubated with FITC-labeled Pep/Gal-PNPs at pH 6.8 and 7.4 for 2 h and then stained using the anti-ASGPR rabbit pAb (diluted with 5% BSA to 1:50) as primary antibody and Alexa 647-labeled goat anti-rabbit IgG (diluted with 5% BSA to 1:200) as the secondary antibody. The colocalization signals were imaged by CLSM.

### Intracellular signaling pathway studies

LO2 cells were seeded on a 12-well plate and cultured for 2 days. Then the cells were incubated with free insulin and insulin-loaded Pep-Gal/PNPs with the same dose of insulin at 20 nM (5.8 ug mL$^{-1}$) for predetermined time intervals. Afterward, the cells were washed and lysed with RIPA containing protease inhibitors and phosphatase inhibitors. The expression of p-AKT in cells was analyzed by western blot (Supplementary Method 11). The phospho-AKT (Ser473) rabbit mAb (diluted with 5% BSA to 1:1000) was utilized as the primary antibody, and the HRP-conjugated goat anti-rabbit IgG (diluted with 5% BSA to 1:5000) as the secondary antibody for the detection of p-AKT in cells.

### Intestinal absorption studies

To directly observe the real-time intestinal absorption of nanoparticles in the living rats, the TPM was performed for intravital imaging as reported previously[32]. The rats were fasted overnight before studies and then injected intraperitoneally with Hoechst 33258 (2 mg kg$^{-1}$). After 30 min, the rats were anesthetized, and the small intestine was gently pulled and stuck to the glass slide. The intestinal segment was cut along one side, and the FITC-labeled nanoparticles at pH 6.8 and 7.4 were added, respectively. Subsequently, the intestinal absorption of nanoparticles along with time was detected using the TPM (Olympus, FV1200MPE, Japan). The integrated densities of images were quantified using ImageJ software (NIH, USA).

For CLSM observation, after the rats were anesthetized, about 5 cm segments of the small intestine were ligated at both ends. Then the FITC/RITC-labeled nanoparticles at pH 6.8 and 7.4 were injected into the loops. After treatment for 2 h, the rats were sacrificed, and the intestinal loops were excised and fixed in 4% paraformaldehyde for 4 h, then stored in 30% sucrose overnight. Afterward, the frozen sections of each loop were obtained using a cryostat (CM1950, Leica, Germany) and then stained with DAPI for 10 min. The absorption of nanoparticles in the intestinal villi was observed by CLSM[53].

### Biodistribution studies

The rats were fasted overnight and then administered orally with PBS or FITC-labeled nanoparticles. After treatment for 4 h, the rats were sacrificed, and the major organs were isolated and examined using the IVIS spectrum system (Perkin Elmer, USA). Additionally, the organs were further sheared by a high-speed disperser (Ultra-Turrax T 25, IKA Werke, Germany), and the fluorescence intensity of tissue homogenates was detected using the microplate reader.

### Liver-targeting ability studies

To detect the liver targeting ability of the nanoparticles, immunofluorescent staining was performed on the liver sections. In brief, the rats were fasted overnight before the study and then administered orally with FITC-labeled nanoparticles. After treatment for 4 h, the rats were sacrificed, and the livers were isolated. Then, the frozen sections of the liver were obtained using a cryostat and stained using anti-ASGPR rabbit pAb as the primary antibody, and Alexa 647 labeled goat anti-rabbit IgG as the secondary antibody. Afterward, the liver sections were stained with DAPI, and the colocalization signals of nanoparticles with ASGPRs were observed by CLSM. The colocalization coefficient (R) was quantified using Imaris software (Bitplane AG, Switzerland).

### Analysis of ligand-switching features of Pep on Pep/Gal-PNPs

FAM and TAMRA (a FRET pair) were conjugated to the N- and C-termini of Pep, respectively, to prepare FR-labeled Pep. Then the FR-labeled Pep was used to prepare Pep/Gal-PNPs (FR-Pep/Gal-PNPs) as described above. The rats were fasted overnight and then orally administered with FR-Pep/Gal-PNPs at a dose of 50 mg kg$^{-1}$. The rats were sacrificed at either 2 h after treatment to collect a 2–3 cm segment of the duodenum or at 4 h to collect a lobe of the liver. After staining with Hoechst 33258 for 15 min, the tissues were observed by CLSM (TCS SP8, Leica, Germany), and the FRET efficiency was analyzed using the FRET acceptor photobleaching method.

### Studies on the systemic delivery route of Pep/Gal-PNPs

The rat was fasted overnight and then orally administered FITC-labeled Pep/Gal-PNPs. After treatment for 2 h, the rat was anesthetized, the abdomen was exposed, and the intestine was externalized for scanning by confocal laser endomicroscopy (ViewnVivo B30, OptiScan, Australia) with a z-step size of 3 μm. Subsequently, the abdominal incision in the rat was sutured. After treatment for an additional 2 h, the liver was scanned following the same procedure.

### Therapeutic efficacy studies on diabetic rats

The diabetic rats were fasted overnight before studies and then administered with different formulations (each group $n = 6$): free insulin solution at a dose of 5 IU kg$^{-1}$ via subcutaneous injection; free insulin solution, insulin-loaded nanoparticles at a dose of 75 IU kg$^{-1}$ via oral gavage. The blood samples were collected from the tail veins of rats before administration and at predetermined time intervals after dosing. The blood glucose level was measured using the glucose meter (On Call® EZ, Acon Biotechnology).

To analyze peripheral serum insulin levels, the blood samples of rats were collected from the eye veins before administration and at predetermined time intervals after dosing. Then the blood samples were centrifuged at 1503 × g for 10 min, and the collected serum was incubated with acetonitrile for 30 min to release the encapsulated insulin. The serum insulin concentrations were determined using a human insulin ELISA kit. The pharmacological availability (PA%) and bioavailability (F%) of nanoparticles relative to subcutaneous injection of insulin were calculated according to the following equations:

$$PA(\%) = \frac{AAC_{oral} \times Dose_{s.c.}}{AAC_{s.c.} \times Dose_{oral}} \times 100 \tag{4}$$

$$F(\%) = \frac{AUC_{oral} \times Dose_{s.c.}}{AUC_{s.c.} \times Dose_{oral}} \times 100 \tag{5}$$

where AAC denotes the area above the blood glucose level versus the time curve.

For the analysis of the portal serum insulin level, the blood samples were collected from the portal vein of rats by cannulation before administration and at predetermined time intervals after dosing. Then the blood samples were centrifuged at 1503 × g for 10 min, and the

collected serum was incubated with acetonitrile for 30 min to release the encapsulated insulin. Then, the portal insulin concentrations were determined using a human insulin ELISA kit.

## Hepatic glycogen measurement
The rats fasted overnight before studies, and then the diabetic rats were subcutaneously injected with free insulin solution (5 IU kg$^{-1}$) or administered orally with insulin-loaded nanoparticles (75 IU kg$^{-1}$). The normal and diabetic rats administered orally with PBS were taken as the positive and negative control, respectively. After treatment for 4 h, the rats were fed food. The rats were daily dosed for one week, and on the last day, the rats were fed food for 2 h and then sacrificed to collect the livers. The hepatic glycogen contents in rats were measured by the glycogen assay kit. Afterward, the livers were fixed with 4% paraformaldehyde and stained using the periodic acid-Schiff (PAS) staining method. The synthesized hepatic glycogen was observed by a light microscope (DM 6B, Leica, Germany).

## In vivo toxicity analysis
The biocompatibility of nanoparticles was investigated following healthy rats being administrated orally with PBS and nanoparticles (1 mg mL$^{-1}$) every day for a week. The body weight of rats was recorded each day after dosing. Meanwhile, the blood samples of rats were collected from eye veins at 0 and 7th day of dosing, and then the serum ALT and AST levels were determined using commercial kits. The rats were sacrificed post dosing, and the livers and small intestines were isolated. Afterward, the organs were fixed with 4% paraformaldehyde, embedded in paraffin, and cut for sections. After staining with hematoxylin and eosin, the histomorphology changes of organs were observed using the light microscope.

## Statistics and reproducibility
All experiments were performed in triplicate unless otherwise stated. A representative image of three replicates from each group was shown. The results were presented as mean ± standard deviation (SD). Two-tailed Student's *t*-test or analysis of variance (ANOVA) was conducted to compare experimental groups in GraphPad Prism 7.0 software. The differences were considered statistically significant for *p* values < 0.05. The levels of significance were set at the probability of *$p$ < 0.05, **$p$ < 0.01, and ***$p$ < 0.001.

## Reporting summary
Further information on research design is available in the Nature Research Reporting Summary linked to this article.

## Data availability
Source data are available for Figs. 3a, b, d, 4b, d, i, 5b, c, 6b–d, f–h, Supplementary Figs. 3, 5–8, 10–17, 19, 21–22, 24–26 and Supplementary Table S1 in the associated "Source Data" file. All the other data that support the findings of this study are available within the Article and its Supplementary Information files and from the corresponding author upon reasonable request. Source data are provided with this paper.

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

## Acknowledgements

We are grateful for the financial support from the National Science Fund of Distinguished Young Scholars (82025032 (Y.G.)), the National Natural Science Foundation of China (82003678 (C.Z.) and 82073773 (M.Y.)), the Major International Joint Research Project of Chinese Academy of Sciences (153631KYSB20190020 (Y.G.)), the Fudan-SIMM Joint Research Fund (FU-SIMM 20173006 (Y.G.)), the Chinese Pharmacopoeia Commission (2021Y20 (S.G.)). We appreciate the technical support in atomic force microscopy (AFM) from the Instrumental Analysis Center of Shanghai Jiao Tong University. We thank Fengming Liu of the Integrated Laser Microscopy System at the National Facility for Protein Science in Shanghai (NFPS, ZJLab) for her help in collecting two-photon microscopy and Leica TCS SP8 confocal microscopy data. We thank the staff members of Electron Microscopy System at the National Facility for Protein Science in Shanghai for the Cryo-TEM data collection and analysis. We also thank Biotimes Technology Co., Ltd for our confocal laser endomicroscopy data. Parts of Figs. 1, 4j, and 6a were created using templates from Servier Medical Art (http://smart.servier.com/), licensed under a Creative Common Attribution 3.0 Generic License.

## Author contributions

T.Y., W.F., G.W., and Y.G. designed this project. T.Y. and X.J. performed all experiments and collected and analyzed the data. T.Y., D.N., and A.W. wrote the manuscript. T.Y., A.W., D.N., W.F., M.Y., S.G., C.Z., G.W., and Y.G. contributed to reviewing the manuscript and discussing the results and implications.

## Competing interests

The authors declare no competing interests.
