## [Peer Review File · Nature Communications]

Ligand-switchable nanoparticles resembling viral surface for sequential drug delivery and improved oral insulin therapyReviewers' comments:

Reviewer #1 (Remarks to the Author):

In this manuscript, the authors presented a switchable ligand-functionalized nanoparticles for the oral delivery of insulin. The overall study and experimental design are well organized and performed. However, the proposal in the manuscript is not sufficiently innovative to be published in Nature Communications. The dual ligands-modified PLGA particles have already been reported elsewhere. The pH-responsiveness of cell-penetrating peptide have long been applied in the construction of drug delivery systems. The concept of virus surface-inspired function is not that convincing since the major function and mechanism of virus surface-mediated cargo transportation is not reflected in the proposed design. Overall, I do not recommend acceptance of this manuscript. It could be suitable for a more specialized journal after addressing following issues.

1. Since the oral drug delivery needs to enter stomach before reach intestine, how stable could Pep/Gal-PNPs be in a more acidic environment (pH=2) like stomach?
2. The accumulation of designed particles in different organs should be statistically analyzed to reflect the in vivo distribution.
3. In Line 45, Page 3, "...remain..." should be "...remains..."

Reviewer #2 (Remarks to the Author):

This study uses a ligand-switchable nanoparticle with a pH-responsive stretchable cell-penetrating peptide and a liver-targeting moiety. This approach was used to allow encapsulated insulin to traverse the intestinal barriers and then specifically target the liver once in the bloodstream. With oral delivery, insulin bioavailability in the circulation was >10%, which is a difficult feat. With regard to the use of this approach for the treatment of diabetes, much remains to be done, but in demonstrating proof of concept they succeed. A major concern is the need to demonstrate insulin dose equivalency with subcutaneous vs Pep/Gal-PNP delivery.

Specific comments:

Lines 24/76 What is a PNP?

40 What is RGD?

85 Fig 1 depicts insulin located within the interior of the nanoparticle, but how this encapsulation occurs is not well explained. What causes insulin to be loaded into a nanoparticle? Also, please provide further explanation of the meanings of entrapment efficiency and loading capacity (186).

182/215/251 "Hemolysis" normally refer to breakdown of RBCs. Please describe the hemolysis assay in the methods.

212 Providing the rationale for using the cell-penetrating segment of Pep as a comparator would be helpful to the reader.

338-339 Please explain why colocalization of FITC and RITC indicates structural integrity of the particles.

Table S1: It is not clear why particle size is not provided for PNP, Pep-PNP, and Pep/Gal-PNP.

408-413 It is difficult to compare the hypoglycemic effect of the nanoparticles compared to free insulin without a way to equate treatment doses of insulin. This is also true for liver glycogen accumulation. Were the delivery doses of 5 vs 75 U/kg chosen based on the relative bioavailability of Pep/Gal-PNP vs sc? This should be noted.

Bioavailability of peripheral insulin (Table 1) was calculated from the AUC of serum insulin, although with this approach bioavailability is likely to be underestimated with respect to hepatocyte insulin exposure, which was possibly higher due to sequestration of nanoparticles to the liver. In addition, hepatic insulin extraction is typically about 50%. Since subcutaneous insulin enters the circulation downstream of the liver, but the nanoparticles were delivered via the portal vein, it can be estimated that the liver was exposed to approximately two-fold more insulin than what was measured in peripheral serum. I.e. with oral vs sc delivery, peripheral and hepatic insulin bioavailability are not the same.

Does the insulin assay measure nanoparticle encapsulated insulin? I.e. are the insulin levels presented in Fig 6 and 24, Tables 1 and S3 "free" or total insulin? If the assay only measures free insulin, why are portal insulin levels higher than peripheral in the Pep/Gal-PNP group?

It is very troubling that portal insulin levels are higher than peripheral in the subcutaneous group. Physiologically, this is not possible, because about 20% of insulin in the blood is degraded as it passes across the gut, therefore with peripheral (sc) delivery peripheral insulin levels are always higher than portal.

Although the PK results appear to show no remaining insulin in the circulation after 12h, the 12h PD effect on blood glucose remains nearly maximal (Fig 6b-c). Is there an explanation for this? The time course for return to basal blood glucose should be shown.

Because % change instead of actual blood glucose levels are provided, and because the rats were diabetic, it is difficult to predict how much blood glucose levels would have been influenced by hypoglycemic counterregulation (i.e. did glucose levels fall below the euglycemic level of a non-diabetic animal, and if so, by how much?). If hypoglycemia indeed occurred, then the PD data are difficult to interpret, since differing degrees of hypoglycemia will elicit differing anti-insulin responses, which will hide the PD effects of insulin. Euglycemic clamps with glucose infusion rates used as the readout would be preferable, although in my opinion this is beyond the scope of a proof of principle study such as this.

Potential hypoglycemia is also an issue for the liver glycogen data. Hypoglycemia will override the effect of insulin on glycogen synthesis, due to counterregulatory hormone stimulation of glycogen breakdown. If subcutaneous insulin caused deeper hypoglycemia (as appears to be the case in Fig 6b, where small differences in blood glucose have a major effect on the strength of hypoglycemic counterregulation response), then this could explain why there was not as much glycogen accumulation in the sc group. With that said, if the insulin doses can be equated, decreased hypoglycemic risk with nanoparticle vs subcutaneous delivery is a benefit that can be noted.

808 Was glycogen measured in livers of fed rats? How long after the last treatment and feeding was the sacrifice?

518-519 A strong "hypoglycemic" response is considered a negative attribute of insulin therapies. On the other hand, depending on the dose and the resulting glucose level, it could be argued that Pep/Gal-PNPs might elicit strong "glucose lowering" effects while also minimizing the hypoglycemic risk that would be present with an equivalent dose of sc delivered insulin (because of sc insulin's preferential effect on muscle vs liver glucose uptake). In addition, based on the PD results, there is the potential for sustained glycemic control compared to sc delivery.

534-535 PMID: 28654313 is an example of a previous manuscript that discusses the mechanism by which oral insulin therapy could increase hepatic glycogen synthesis.

538-542 Greater efficacy in promoting liver glycogen storage depends on insulin dose equivalency. The authors need further evidence of dose equivalency to support this claim.

541 Overly hepatoselective insulin will likely result in hepatosteatosis, which has led some hepatoselective insulin analogs to fail. The strength of this manuscript is in the novel drug delivery approach. The insulin results are interesting because they provide proof of principle for the

approach but much remains to be done to demonstrate that this approach will be an effective way to treat diabetes, however. This aspect of the discussion should be tempered.

Response to comments on

“Virus surface-inspired ligand-switchable nanoparticles enable sequential drug delivery for improved oral insulin therapy (NCOMMS-22-11316)”

Reviewer #1:

In this manuscript, the authors presented a switchable ligand-functionalized nanoparticles for the oral delivery of insulin. The overall study and experimental design are well organized and performed. However, the proposal in the manuscript is not sufficiently innovative to be published in Nature Communications. The dual ligands-modified PLGA particles have already been reported elsewhere. The pH-responsiveness of cell-penetrating peptide have long been applied in the construction of drug delivery systems. The concept of virus surface-inspired function is not that convincing since the major function and mechanism of virus surface-mediated cargo transportation is not reflected in the proposed design. Overall, I do not recommend acceptance of this manuscript. It could be suitable for a more specialized journal after addressing following issues.

Responses: Thank you for your comments.

It seems to be a misunderstanding about our work, which we hope to clear through our responses below and by revising the text in the manuscript.

1) The concern about the novelty of our work.

Surface modification of nanoparticles with various types of ligands has the potential to improve drug delivery efficiency. However, mutual interference (e.g., steric hindrance or electrostatic repulsion) between surface ligands could compromise their functions and lead to undesirable outcomes. This issue has not yet been fully addressed in previous publications. In this study, we designed novel dual-ligand nanoparticles (Pep/Gal-PNPs) that consist of pH-responsive cell-penetrating peptide (Pep) and hepatic targeting ligand (Gal). The Pep could change its conformation in response to environmental pH, which adopts a stretched conformation at acidic pH while turns folded at physiological pH. Therefore, following oral administration, Pep/Gal-PNPs expose Pep ligand on the surface to traverse intestinal barriers and then switch to Gal ligand to specifically target the liver. Collectively, Pep/Gal-PNPs could modulate dual surface ligands to expose the specific one at targets, avoiding mutual interference and unleashing their full potential for efficient drug delivery.

Although the pH-responsive CPPs were studied in previous publications (*Adv Drug*

Deliv Rev. 2013, 65(10):1299-1315) as described by the reviewer, we applied them to develop the aforementioned multifunctional nanoparticles to avoid mutual interference between surface ligands. The conformation/length and stretchable structure of the pH-responsive CPPs used in our study (denoted Pep) were precisely designed and thoroughly characterized by atomic force microscope (AFM) and fluorescence resonance energy transfer (FRET) techniques, which had not been reported previously. The results indicated that stretchable changes in the structure of the Pep on Pep/Gal-PNPs enable it to function selectively and expose the Gal at specific sites.

Overall, we believe this is a pioneer work, and the previous publications could not diminish its novelty.

2) The concern about the concept of virus surface-inspired function of our design.

Our design simulates the major surface functions and mechanisms of viruses, i.e., it mimics viruses to modulate surface proteins conformations to sequentially exert multiple functions in a coordinated fashion, avoiding mutual interference between diverse ligands. Influenza A virus (IAV), for example, is modified with two major surface proteins, neuraminidase (NA) and hemagglutinin (HA); The NA mediates the IAV to penetrate mucus barriers, and the HA modulates its conformations to expose fusion segment and help IAV fuse with endosomal membrane after entering cells (*Nature*. 2020, 583(7814):150-153; *Proc Natl Acad Sci U S A*. 2017, 114(42):11157-11162). By virtue of this conformational change, the NA and HA on IAV could exert their respective functions without interference, realizing efficient invasion. Likewise, the Pep/Gal-PNPs designed here are modified with two functionalities (Pep and Gal). The Pep shares similar conformational changes to HA, which mediates Pep/Gal-PNPs to traverse intestinal barriers and then exposes the Gal ligand to target the hepatocytes. Therefore, by mimicking viral unique surface features, the Pep/Gal-PNPs could avoid mutual interference between dual ligands and fully unleash their functions for efficient drug delivery.

We feel sorry that our points are not clear in our original manuscript, leading to misunderstanding by reviewer. We have now revised our manuscript to stress the novelty of our research, and the main modifications in the Introduction and Discussion sections are as follows:

“These findings strongly inform us that through mimicking the unique surface properties of viruses to modulate conformations/structures of surface ligands, the multifunctional nanoparticles are anticipated to diminish mutual interference between

diverse ligands and realize their full potentials for efficient drug delivery.” (Page 4, Line 70-74).

“In summary, the Pep/Gal-PNPs simulate unique viral surface features that modulate the conformations of Pep ligands to fully exert diverse functions of dual surface ligands, avoiding their mutual interferences and ultimately improving oral insulin therapy.” (Page 4, Line 91-94)

“Despite dual-ligand modified vehicles having been widely applied to meet these demands, mutual interference between surface ligands might be an important factor limiting their functionalities and thus resulting in low in vivo drug delivery efficiency^{1, 2}.” (Page 16, Line 495-498)

“Therefore, the Pep/Gal-PNPs share similar mechanisms and functions to viruses that switch dual-functional ligands to avoid their interference via modulating Pep conformations in response to environmental pH.” (Page 16, Line 506-509)

“It is noteworthy that this work presents a promising strategy to fully exert diverse functions of dual surface ligands on multifunctional nanocarriers, which is distinct from previous studies that focus on improving synergetic effects of diverse ligands^{3, 4}.” (Page 17, Line 540-543)

Specific issues:

1. Since the oral drug delivery needs to enter stomach before reach intestine, how stable could Pep/Gal-PNPs be in a more acidic environment (pH=2) like stomach?

Responses: Thank you for raising this issue.

The stability of Pep/Gal-PNPs was investigated in simulated gastric fluid (SGF, pH=1.2 with 1% pepsin) and simulated intestinal fluid (SIF, pH=6.8 with 1% trypsin). As shown in Fig. R1 (Supplementary Fig. 7), the size and polydispersity index (PDI) of Pep/Gal-PNPs remained unchanged, suggesting they were stable in harsh gastrointestinal environments. Meanwhile, the circular dichroism results indicated that the released insulin still retained activity (Supplementary Fig. 8b). Moreover, we have also discussed the results in our manuscript:

“Overall, these results indicated that Pep/Gal-PNPs could remain stable in the harsh gastrointestinal environment ...” (Page 7, Line 201-202)

Fig. R1. *In vitro* stability of Pep/Gal-PNPs. a) The relative size and b) polydispersity intensity (PDI) of Pep/Gal-PNPs in SGF and SIF. Data are presented as the mean \pm SD (n=3).

2. The accumulation of designed particles in different organs should be statistically analyzed to reflect the *in vivo* distribution.

Responses: Thank you for your suggestion.

We have quantified the ratio of nanoparticles that accumulate in the major organs of rats. The results have been shown in our revised manuscript (Supplementary Fig. 21, Fig. R2)

Fig. R2 (Supplementary Fig. 21 in revised manuscript). Quantitative analysis of accumulation ratio of PNP, CPP/Gal-PNP, and Pep/Gal-PNP in major organs. Data are presented as the mean \pm SD (n=3). * $p < 0.05$ and ** $p < 0.01$ compared with the Pep/Gal-PNP group.

Besides, additional discussions were also provided in our revised manuscript:

“The amount of Pep/Gal-PNPs accumulated in the liver and kidney was almost 71.9% \pm 3.7% and 12.5% \pm 1.5% of the total absorption, respectively. In contrast, the ratio of CPP/Gal-PNPs that accumulated in the kidney increased to 23.0% \pm 4.0% (Supplementary Fig. 21).” (Page 12, Line 359-362)

3. In Line 45, Page 3, “...remain...” should be “...remains...”.

Responses: Thank you for pointing out this error.

We are sorry for the mistake. We have now thoroughly checked and corrected the grammatical errors and typos we found in our revised manuscript.

Reviewer #2:

This study uses a ligand-switchable nanoparticle with a pH-responsive stretchable cell-penetrating peptide and a liver-targeting moiety. This approach was used to allow encapsulated insulin to traverse the intestinal barriers and then specifically target the liver once in the bloodstream. With oral delivery, insulin bioavailability in the circulation was >10%, which is a difficult feat. With regard to the use of this approach for the treatment of diabetes, much remains to be done, but in demonstrating proof of concept they succeed. A major concern is the need to demonstrate insulin dose equivalency with subcutaneous vs Pep/Gal-PNP delivery.

Responses: We thank the reviewer for the thoughtful review and suggestive comments on our study.

In our responses, we have mainly addressed the following concerns:

1) The insulin treatment dose selection:

In our study, the insulin treatment doses for subcutaneous insulin (s.c. INS) and Pep/Gal-PNPs groups were selected to achieve comparable peripheral insulin exposure (reflected by AUC), thus ensuring their similar glucose deposition in the periphery. On this premise, we further compared their hepatic insulin exposure and corresponding hepatic glycogen production. The detailed responses to this issue (questions 8, 14, 16, and 18) are listed below;

2) The concern about the hypoglycemic counterregulation might influence the pharmacodynamic and hepatic glycogen production results in s.c. INS group:

It could not affect our results since numerous studies have demonstrated that the key counterregulatory hormone responses are lost during subcutaneous insulin-induced hypoglycemia in type I diabetes, i.e., the hepatic glycogen breakdown and glucose production are suppressed in this condition (J Clin Invest. 2016, 126(6):2236-2248; Diabetes. 2015, 64(10):3439-3451; N Engl J Med. 1985, 313(4):232-241; J Clin Invest. 1984, 73(6):1532-1541). Therefore, the hypoglycemia in s.c. INS group could not influence their hypoglycemia performance as well as hepatic glycogen data. The detailed responses to this issue (questions 13 and 14) are listed below;

3) Responses to other specific issues are also listed below.

Specific issues:

1. Lines 24/76 What is a PNP?

Responses: Thank you for raising this issue.

We have added the full name of PNP in our revised manuscript:

“We develop ligand-switchable poly (lactic-co-glycolic acid) (PLGA) nanoparticles (PNP) ...” (Page 4, Line 77-78)

2. 40 What is RGD?

Responses: Thank you for raising this issue.

We have added the full name of RGD in our revised manuscript:

“For example, the covalent modification with tripeptide Arg-Gly-Asp (RGD) peptides and transferrin (Tf) ligands ...” (Page 3, Line 40)

3. 85 Fig 1 depicts insulin located within the interior of the nanoparticle, but how this encapsulation occurs is not well explained. What causes insulin to be loaded into a nanoparticle? Also, please provide further explanation of the meanings of entrapment efficiency and loading capacity (186).

Responses: Thank you for pointing out this issue.

Here, the double emulsion and solvent evaporation method has been applied for preparing the insulin-loaded nanoparticles, as reported previously (*Nat Commun.* 2021, 12(1):2935; *J Control Release.* 2019, 295:31-49). The insulin is first emulsified with the polymer solution to form the primary emulsion (W1/O) and then emulsified with the outer aqueous phase containing surfactant to form a double emulsion (W1/O/W2). The insulin is entrapped in the internal aqueous phase, and the organic phase is removed by evaporation. The added surfactants could stabilize nanoparticles in water.

We have supplemented the calculation formula of the loading capacity and entrapment efficiency of insulin in nanoparticles in the Materials and Methods section of our revised manuscript:

“The nanoparticles were demulsified using acetonitrile, then the entrapment efficiency (EE) and loading capacity (LC) of insulin in nanoparticles were quantified by high-performance liquid chromatography (HPLC; Agilent 1260, USA) and calculated using the following equation:

$$\text{Loading capacity (LC\%)} = \frac{\text{Total amount of insulin loaded} - \text{free insulin}}{\text{Total amount of insulin}} \quad (1)$$

$$\text{Entrapment efficiency (EE\%)} = \frac{\text{Total amount of insulin loaded} - \text{free insulin}}{\text{Nanoparticles weight}} \quad (2)''$$

(Page 22, Line 691-696)

4. 182/215/251 “Hemolysis” normally refer to breakdown of RBCs. Please describe the hemolysis assay in the methods.

Responses: Thank you for your advice.

The hemolysis studies were performed on red blood cells (RBCs) of rats, and the method was described in Supporting Information (Supplementary Note 3):

“Supplementary Note 3. Hemolytic effect test.

The rat blood was collected and then centrifuged at 4000 rpm for 10 min. The red blood cells (RBCs) were obtained and washed three times with fresh PBS. Then the RBCs were diluted with PBS at pH 6.0, 6.5, 7.0, 7.5, 8.0 respectively to 2% (w/v). The Pep and CPP were also dissolved in PBS of different pH to a final concentration of 0.5 mg/mL. Then, 100 μL Pep or CPP solution was added to an equal volume of RBCs suspension and incubated at 37 °C for 1 h. After centrifugation, the supernatants were collected, and the absorbance at 570 nm was measured using a microplate reader. The RBCs that were treated with PBS and 1% Triton X-100 were set as control groups, and the hemolysis level of Pep and Gal was measured using the following formula:

$$\text{Hemolysis}\% = \frac{A(\text{sample}) - A(\text{PBS})}{A(\text{Triton}) - A(\text{PBS})} \times 100 \quad (1)$$

Furthermore, the hemolytic effect of nanoparticles was studied using the method mentioned above, and the concentration was set as 1 mg/mL.” (Page 4 in Supporting Information)

5. 212 Providing the rationale for using the cell-penetrating segment of Pep as a comparator would be helpful to the reader.

Responses: Thank you for your advice.

We have addressed this issue in the Results and Discussion section of our revised manuscript:

“The cell-penetrating segment of Pep (CPP, R₆) was demonstrated to be pH-insensitive (Supplementary Fig. 11). Thus, the non-switchable nanoparticles (CPP/Gal-PNPs) modified with CPP and Gal were developed as a comparator (Supplementary Table S1), aiming to verify the cell-penetrating ability of Pep ligand and ascertain the advantages of ligand-switchable features of Pep/Gal-PNPs. Moreover, CPP/Gal-PNPs showed potent, pH-insensitive hemolytic activity (Supplementary Fig. 11).” (Page 8, Line 217-223)

“In contrast, although CPP/Gal-PNPs modified with the cell-penetrating segment of Pep exhibited similar enhanced intestinal absorption efficiency, they lacked cell specificity in the liver and increased the accumulation ratios in other organs, which was attributed to non-switchable surface ligands.” (Page 17, Line 534-538)

6. 338-339 Please explain why colocalization of FITC and RITC indicates structural integrity of the particles.

Responses: Thank you for pointing out this issue.

The FITC and RITC were simultaneously encapsulated into the nanoparticles, and their colocalization signals appear yellow in confocal laser scanning microscopy (CLSM) images. If the nanoparticles disassemble *in vivo*, these fluorescence molecules are released and change their location individually, and the colocalization signals can not be detected. Therefore, the strong colocalization signals observed in CLSM images of intestinal slices suggested the structural integrity of nanoparticles (Figure 5a, Fig. R3). Moreover, the dual-labeling method has been widely applied to investigate the structural integrity of nanoparticles in previous publications (*Adv. Funct. Mater.* 2020, 30(13):1910168; *Nat Nanotechnol.* 2015, 10(7):619-623).

Fig. R3. Two-photon microscopy (TPM) images show the absorption of nanoparticles in intestinal villi. Scale bar: 100 μm . Confocal laser scanning microscopy (CLSM) images of intestinal villus sections. Scale bars: intestinal slice images, 200 μm ; enlarged images, 50 μm .

7. Table S1: It is not clear why particle size is not provided for PNP, Pep-PNP, and

Pep/Gal-PNP.

Responses: Thank you for pointing out this issue.

The size and zeta potential of PNP, Pep-PNP, and Pep/Gal-PNP have been shown in our manuscript. To avoid misunderstanding, we have supplemented that in Supplementary Information (Table S1, Table R1).

Table R1 (Supplementary Table S1 in revised manuscript). Characterization of nanoparticles. Data are presented as the mean \pm SD (n=3).

Group	Size (nm)	Zeta Potential (mV)	EE ^a (%)	LC ^b (%)	Pep/CPP modification rate (%)	Gal modification rate (%)
PNP	108.6 \pm 1.9 (pH 6.8)	-33.8 \pm 2.3 (pH 6.8)	46.3	7.5	-	-
	107.2 \pm 3.6 (pH 7.4)	-37.4 \pm 2.9 (pH 7.4)				
Gal-PNP	129.1 \pm 4.3 (pH 6.8)	-33.6 \pm 2.6 (pH 6.8)	53.8	8.4	-	5.2
	128.0 \pm 2.7 (pH 7.4)	-35.4 \pm 2.5 (pH 7.4)				
Pep-PNP	135.0 \pm 3.8 (pH 6.8)	24.7 \pm 2.1 (pH 6.8)	42.1	6.8	4.7	-
	117.6 \pm 4.0 (pH 7.4)	-30.8 \pm 1.4 (pH 7.4)				
CPP/Gal-PNP	126.8 \pm 2.4 (pH 6.8)	28.3 \pm 1.7 (pH 6.8)	44.5	7.2	4.9	5.8
	126.2 \pm 3.9 (pH 7.4)	27.4 \pm 2.4 (pH 7.4)				
Pep/Gal-PNP	136.1 \pm 3.1 (pH 6.8)	25.1 \pm 3.3 (pH 6.8)	48.1	7.9	5.4	5.6
	126.5 \pm 1.1 (pH 7.4)	-28.2 \pm 3.9 (pH 7.4)				

^a EE: Entrapment efficiency; ^b LC: Loading capacity.

8. 408-413 It is difficult to compare the hypoglycemic effect of the nanoparticles compared to free insulin without a way to equate treatment doses of insulin. This is also true for liver glycogen accumulation. Were the delivery doses of 5 vs 75 U/kg chosen based on the relative bioavailability of Pep/Gal-PNP vs sc? This should be noted.

Responses: Thank you for raising these issues.

In our study, the insulin treatment doses for s.c. INS and Pep/Gal-PNP groups are selected mainly based on their similar peripheral insulin exposure (182.2 ± 7.1 vs. 210.6 ± 14.9 $\mu\text{IU}\cdot\text{h}/\text{mL}$) to ensure they achieve comparable glucose deposition in the periphery. On that basis, we further determine their portal insulin exposure and hepatic glycogen production in diabetic rats. Notably, these groups would generate different insulin concentrations in systemic and hepatic portal circulation by applying equivalent insulin doses, thus making it difficult to compare their contribution to lowering blood glucose and promoting hepatic glycogen production.

Additionally, the insulin treatment doses of 5 IU/kg for subcutaneous insulin administration and 75 IU/kg for oral insulin formulations have also been widely applied in previous publications (*Nat Nanotechnol.* 2020, 15(7):605-614; *Proc Natl Acad Sci U S A.* 2019, 116(12):5362-5369; *ACS Nano.* 2015, 9(3):2345-2356; *Adv. Healthcare Mater.* 2019, 8(12): e1801123).

Collectively, we believe the insulin treatment dose utilized in this study is reasonable.

9. Bioavailability of peripheral insulin (Table 1) was calculated from the AUC of serum insulin, although with this approach bioavailability is likely to be underestimated with respect to hepatocyte insulin exposure, which was possibly higher due to sequestration of nanoparticles to the liver. In addition, hepatic insulin extraction is typically about 50%. Since subcutaneous insulin enters the circulation downstream of the liver, but the nanoparticles were delivered via the portal vein, it can be estimated that the liver was exposed to approximately two-fold more insulin than what was measured in peripheral serum. I.e. with oral vs sc delivery, peripheral and hepatic insulin bioavailability are not the same.

Responses: Thank you for pointing out this issue.

We should address that the 50% clearance rate in the liver is for free insulin rather than Pep/Gal-PNPs. As the Pep/Gal-PNPs could selectively target the hepatocytes, they largely accumulate in the liver (about 71.9% of the total absorption) and serve as an

insulin reservoir to maintain a high hepatic insulin exposure in diabetic rats. Therefore, the AUC of portal serum insulin was significantly higher than that of periphery in Pep/Gal-PNP group.

We agree with the reviewer that there might be a bias in using the peripheral serum insulin concentration to calculate insulin bioavailability. Therefore, we further estimate the hepatic insulin availability (F_h) of Pep/Gal-PNPs relative to subcutaneous injection of insulin by simultaneously measuring the serum insulin level in the eye (systemic) and portal veins in the same rats (responses to Question 11 for details). The F_h is calculated using the following equation according to previous publications (Curr Clin Pharmacol. 2016, 11(1):47-52):

$$F_h = \frac{AUC_{NP,systemic}}{AUC_{S.c.,systemic}} \times \frac{AUC_{S.c.,portal}}{AUC_{NP,portal}}$$

where $AUC_{NP,systemic}$ and $AUC_{S.c.,systemic}$ are the systemic (eye vein) AUCs and $AUC_{NP,portal}$ and $AUC_{S.c.,portal}$ are the portal AUCs following oral administration of Pep/Gal-PNPs and subcutaneous injection of insulin, respectively.

Therefore, the hepatic insulin availability (F_h) of Pep/Gal-PNPs was calculated to be about 19.9%, significantly higher than their peripheral insulin bioavailability (7.7%). To avoid misunderstanding, we have supplemented additional discussion in our revised manuscript:

“Additionally, there might be a bias in using peripheral serum insulin AUC to calculate the oral bioavailability of Pep/Gal-PNPs. Therefore, we further calculated the hepatic insulin availability (F_h) of Pep/Gal-PNPs, and the results indicated it was significantly higher than their peripheral insulin bioavailability (19.9% vs. ~7%) (Supplementary Fig. 26)” (Page 15, Line 463-467)

“The hepatic insulin availability (F_h) of Pep/Gal-PNPs relative to subcutaneous injection of insulin was measured using the following formula:

$$F_h = \frac{AUC_{NP,systemic}}{AUC_{S.c.,systemic}} \times \frac{AUC_{S.c.,portal}}{AUC_{NP,portal}} \quad (2)$$

where $AUC_{NP,systemic}$ and $AUC_{S.c.,systemic}$ are the systemic (eye vein) AUCs and $AUC_{NP,portal}$ and $AUC_{S.c.,portal}$ are the portal AUCs following oral administration of Pep/Gal-PNPs and subcutaneous injection of insulin, respectively.” (Page 8 in Supporting Information)

10. Does the insulin assay measure nanoparticle encapsulated insulin? I.e. are the

insulin levels presented in Fig 6 and 24, Tables 1 and S3 “free” or total insulin? If the assay only measures free insulin, why are portal insulin levels higher than peripheral in the Pep/Gal-PNP group?

Responses: Thank you for raising this issue.

We feel sorry that we did not provide enough details regarding the methods of this measurement. Our studies measured the total insulin levels to evaluate the pharmacokinetic parameters of different insulin formulations. The intact Pep/Gal-PNPs could traverse intestinal barriers and primarily accumulate in the liver (about 71.9% of the total absorption). Then, the encapsulated insulin got released from Pep/Gal-PNPs to take effect on hepatocytes. As the hepatic insulin clearance rate is fast, the half-life of insulin in the portal circulation is about 3~5 min (Physiology (Bethesda). 2019, 34(3):198-215). Therefore, insulin mainly exists in the encapsulated form in the liver. Moreover, the peripheral insulin is in the free form, which is derived from insulin released by Pep/Gal-PNPs in the liver. Accordingly, in Pep/Gal-PNP group, portal serum insulin levels are higher than peripheral. To avoid misunderstanding, we have added the details in the Materials and Methods section of our revised manuscript:

“To analyze peripheral serum insulin levels, ... Then the blood samples were centrifuged at 4000 rpm for 10 min, and the collected serum was incubated with acetonitrile for 30 min to release the encapsulated insulin.” (Page 26, Line 829-831)

“For the analysis of the portal serum insulin level, ... Then the blood samples were centrifuged at 4000 rpm for 10 min, and the collected serum was incubated with acetonitrile for 30 min to release the encapsulated insulin.” (Page 26, Line 840-842)

11. It is very troubling that portal insulin levels are higher than peripheral in the subcutaneous group. Physiologically, this is not possible, because about 20% of insulin in the blood is degraded as it passes across the gut, therefore with peripheral (sc) delivery peripheral insulin levels are always higher than portal.

Responses: Thank you for pointing out this issue.

The portal insulin and peripheral insulin levels were determined based on different purposes, and thus these two experiments were performed separately with different batches of rats. The peripheral serum insulin level was measured to reflect systemic insulin exposure. The portal serum insulin level was determined to illustrate the contribution to promoting hepatic glycogen production. Therefore, we speculate that this issue may be attributed to the high variability between various batches of rats.

We have supplemented the experiments in which the peripheral and hepatic blood samples were simultaneously collected from the same rat after dosing. The result indicated that Pep/Gal-PNP group exhibited higher insulin levels in the portal vein than periphery (1179.0 ± 62.5 vs. 283.8 ± 15.5 $\mu\text{IU}\cdot\text{h}/\text{mL}$), producing a similar liver-to-periphery insulin gradient (almost 4.2:1) to normal subjects (*Diabetes*, 2014, 63(5):1445-1447). In contrast, this gradient was eliminated in s.c. INS and peripheral serum insulin AUC was about 1.2-fold higher than that of portal vein (256.1 ± 8.5 vs. 211.8 ± 7.1 $\mu\text{IU}\cdot\text{h}/\text{mL}$) (Supplementary Fig. 26 in revised manuscript, Fig. R4). It is noteworthy that Pep/Gal-PNP group still exhibited higher hepatic insulin exposure compared to s.c. INS group, which could not affect our previous conclusions.

Fig. R4. (Supplementary Fig. 26 in revised manuscript). Simultaneous measurement of portal (Po) and peripheral (Pe) a) serum insulin level versus time profiles and b) the total area under the serum insulin level versus time curve in type I diabetic rats following oral administration of insulin-loaded Pep/Gal-PNP (75 IU/kg) and subcutaneous injection of insulin (5 IU/kg). Data are presented as the mean \pm SD (n=6).

Moreover, to avoid misunderstanding, we have supplemented this experiment and provided additional discussions in our revised manuscript:

“Supplementary Note 13. Hepatic insulin availability studies of Pep/Gal-PNPs.

The diabetic rats were fasted overnight before studies and then administered with free insulin solution at a dose of 5 IU/kg via subcutaneous injection and insulin-loaded Pep/Gal-PNPs at a dose of 75 IU/kg via oral gavage. The blood samples were collected simultaneously from the eye vein and portal vein of rats before administration and at predetermined time intervals after dosing. Then the blood samples were centrifuged at

4000 rpm for 10 min, and the collected serum was incubated with acetonitrile for 30 min to release the encapsulated insulin. The serum insulin concentrations were determined using a human insulin ELISA kit.” (Page 7 in Supplementary Information)

“Moreover, through simultaneously analyzing the peripheral and portal serum insulin levels on the same rats, the results indicated Pep/Gal-PNPs produced a ~4.2-fold increase in insulin level exposed to the liver than periphery (Supplementary Fig. 26), mimicking the biodistribution of endogenous insulin⁵. In contrast, this insulin gradient was lost in rats following subcutaneous insulin administration, which exhibited higher insulin concentration in the peripheral circulation and might cause hypoglycemia events.” (Page 15, Line 460-466)

12. Although the PK results appear to show no remaining insulin in the circulation after 12h, the 12h PD effect on blood glucose remains nearly maximal (Fig 6b-c). Is there an explanation for this? The time course for return to basal blood glucose should be shown.

Responses: Thank you for pointing out this interesting question.

The temporal separation between the PK and PD profiles of insulin is the consequence of a series of insulin-specific phenomena, including the fraction and rate of absorption, the rate of binding to insulin receptors, subsequent intracellular downstream signaling transduction, and induction of metabolic processes. The kinetics of these events occur over an extended time relative to insulin availability (*Int J Clin Pract. 2010, 64(10):1415-1424; J Diabetes Sci Technol. 2014; 8(4):821–829; Diabetes Res. Clin. Pract. 2019, 148:93-101*). Therefore, it could explain the lag time in PD compared to PK profiles which is a common phenomenon in insulin-related studies (Fig. R5).

We feel sorry that the time course for the return to basal blood glucose in PD studies is not available at present. Because the rats fasted 12 h before the study, and for animal welfare, one-time fasting period should be no longer than 24 h according to IACUC guidelines. Therefore, only 12-h PD and PK results could be obtained. We speculate that the blood glucose of rats in Pep/Gal-PNP group would return to the basal level in the following 4-5 h.

Fig. R5. Comparison of pharmacokinetics (PK) and pharmacodynamics (PD) over time after a single subcutaneous injection of insulin (*Int J Clin Pract.* 2010, 64(10):1415-1424).

13. Because % change instead of actual blood glucose levels are provided, and because the rats were diabetic, it is difficult to predict how much blood glucose levels would have been influenced by hypoglycemic counterregulation (i.e. did glucose levels fall below the euglycemic level of a non-diabetic animal, and if so, by how much?). If hypoglycemia indeed occurred, then the PD data are difficult to interpret, since differing degrees of hypoglycemia will elicit differing anti-insulin responses, which will hide the PD effects of insulin. Euglycemic clamps with glucose infusion rates used as the readout would be preferable, although in my opinion this is beyond the scope of a proof of principle study such as this.

Responses: Thank you for raising this issue.

The actual blood glucose level (BGL) of rats has been provided (Supplementary Fig. 24), which shows that the s.c. INS group reaches the minimum of 65.1 ± 7.7 mg/dL at 3 h after dosing. However, the glucose counterregulatory responses are blunted in diabetes, failing to increase hepatic glucose production in response to hypoglycemia (*J Clin Invest.* 2016, 126(6):2236-2248). Moreover, in our study, the duration of hypoglycemia in diabetic rats was short, no more than half an hour (Figure 6b). Therefore, the mild and short-term hypoglycemia that occurred in diabetic rats that received subcutaneous insulin could not significantly influence their PD results.

The euglycemic clamp is often applied to detect insulin sensitivity in type 2 diabetes models. However, we established the rat model of type 1 diabetes in the study to investigate the hypoglycemic effects of our design. Therefore, the euglycemic clamp is not suitable for our experimental purposes.

14. Potential hypoglycemia is also an issue for the liver glycogen data. Hypoglycemia will override the effect of insulin on glycogen synthesis, due to counterregulatory hormone stimulation of glycogen breakdown. If subcutaneous insulin caused deeper hypoglycemia (as appears to be the case in Fig 6b, where small differences in blood glucose have a major effect on the strength of hypoglycemic counterregulation response), then this could explain why there was not as much glycogen accumulation in the sc group. With that said, if the insulin doses can be equated, decreased hypoglycemic risk with nanoparticle vs subcutaneous delivery is a benefit that can be noted.

Responses: Thank you for bringing up this interesting issue.

Our results demonstrated that elevated hepatic glycogen production in Pep/Gal-PNP group was attributed to its higher hepatic insulin exposure compared with s.c. INS group. In contrast, s.c. INS group shifts the primary site of insulin action away from the liver to peripheral tissues (such as skeletal muscle), leading to insufficient hepatic glycogen production. However, the glucose counterregulation is blunted in diabetes, failing to compensate for the subcutaneous insulin-induced hypoglycemia. Therefore, the hypoglycemia in s.c. INS group would not significantly impact the results of hepatic glycogen synthesis. Moreover, we should stress that the hepatic glycogen content in s.c. INS group is comparable to the results in previous studies (*Adv Funct Mater.* 2020, 30(13):1910168).

15. 808 Was glycogen measured in livers of fed rats? How long after the last treatment and feeding was the sacrifice?

Responses: Thank you for pointing out this issue.

In the hepatic glycogen measurement studies, the rats were fed food after treatment with different formulations for 4 h. Moreover, hepatic glycogen levels in normal and diabetic rats fed daily food were also measured (Figure 6d-e). On the last day of dosing, after treatment with different formulations for 4 h, the rats were fed food for 2 h and then sacrificed to detect their hepatic glycogen contents. To avoid misunderstanding, we have added details in the Materials and Methods section of our revised manuscript:

“After treatment for 4 h, the rats were fed food. The rats were daily dosed for one week, and on the last day, the rats were fed food for 2 h and then sacrificed to collect the livers.” (Page 27, Line 848-850)

16. 518-519 A strong “hypoglycemic” response is considered a negative attribute of insulin therapies. On the other hand, depending on the dose and the resulting glucose level, it could be argued that Pep/Gal-PNPs might elicit strong “glucose lowering” effects while also minimizing the hypoglycemic risk that would be present with an equivalent dose of sc delivered insulin (because of sc insulin’s preferential effect on muscle vs liver glucose uptake). In addition, based on the PD results, there is the potential for sustained glycemic control compared to sc delivery.

Responses: Thank you for raising this interesting issue.

The hypoglycemic effect of Pep/Gal-PNPs is slow but sustained, and the BGL of diabetic rats was decreased to the normal range (about 70% of initial) at 6 h after dosing. It has been demonstrated that directly delivering insulin to the liver could lessen hypoglycemia (*Diabetes*, 2015, 64:3439–3451). Moreover, it is noteworthy that the hypoglycemic response elicited by Pep/Gal-PNPs in our study is comparable to other oral insulin formulations reported in previous publications. For example, Xiao et al. designed a glucose-responsive oral insulin nanoparticle (Ins/NP-Fc) which reduced BGL to normal levels within 4 h (*Matter*. 2021, 4(10):3269-3285); Han et al. developed zwitterionic micelles for oral insulin delivery, and PD results indicated that about 70% reduction in BGL at 6 h after treatment (*Nat. Nanotechnol.* 2020, 15(7):605-614); The lipid nanovesicles designed by Yu et al. showed a maximal glucose reduction (~50% of initial) at 4 h after dosing (*Proc. Natl. Acad. Sci. U. S. A.* 2019, 116(12):5362-5369). Therefore, we believe the glucose-lowering effect of the Pep/Gal-PNPs is reasonable and acceptable.

17. 534-535 PMID: 28654313 is an example of a previous manuscript that discusses the mechanism by which oral insulin therapy could increase hepatic glycogen synthesis.

Responses: Thank you for providing this reference.

This paper (PMID 28654313) has been cited in our previous manuscript (Ref. 45) to illustrate the importance of controlling glycemic stability for diabetes.

On the one hand, the reviewer might think it is not appropriate to claim that “... these findings are the first attempt to explicit the mechanism for the liver-targeting oral insulin therapy” (Page 17, Line 534-535). In our study, we investigate the hepatic glycogen synthesis mechanism elicited by the oral insulin nanoparticles with liver selectivity by studying the intracellular signaling pathway, which has not been reported

to the best of our knowledge. This paper (PMID 28654313) provides the principle that the oral insulin delivered to the liver has advantages in managing hepatic glucose production, which is totally different from our study. To avoid the confusing statement, we have revised the related sentences in our revised manuscript:

“Although oral insulin therapy is generally considered to have advantages in promoting hepatic glycogen synthesis⁶, these findings explicit the intracellular mechanism behind that elicited by our liver-targeting oral insulin therapy.” (Page 18, Line 566-569)

On the other hand, the reviewer might want us to explain the differences between our study and previous oral insulin therapy on promoting hepatic glycogen synthesis. Although oral administration shows advantages over other administration routes in promoting hepatic glycogen synthesis for insulin therapy, the major hurdles for oral insulin therapy are how to deliver insulin to transport across formidable gastrointestinal barriers and subsequent target the liver. However, most studies only focus on addressing the former question. In our study, we designed multifunctional nanoparticles that could meet both demands, ultimately specifically targeting insulin to the aimed sites (hepatocytes) in the liver. Therefore, our study is distinct from previous publications and may provide a reference for the innovation of oral insulin therapy.

18. 538-542 Greater efficacy in promoting liver glycogen storage depends on insulin dose equivalency. The authors need further evidence of dose equivalency to support this claim.

Responses: Thank you for your advice.

As we mentioned above, insulin dose equivalency for s.c. INS and Pep/Gal-PNP groups increased the difficulty of comparing their effects on promoting hepatic glycogen production. Therefore, the doses were selected based on their similar peripheral insulin exposure. Under this premise, our studies demonstrated that Pep/Gal-PNPs could significantly increase liver glycogen production by improving hepatic insulin exposure. Moreover, the aim of this work is to develop novel multifunctional nanoparticles to unleash the full potential of diverse surface ligands, and insulin is chosen as the model drug for the proof-of-concept study on the therapeutic potential of our design. The *in vivo* therapeutic studies have verified that Pep/Gal-PNPs exhibited optimal hypoglycemic effects and effectively promoted hepatic glycogen synthesis. Collectively, dose equivalency experiments are beyond the scope of our study.

19. 541 Overly hepatoselective insulin will likely result in hepatosteatosis, which has led some hepatoselective insulin analogs to fail. The strength of this manuscript is in the novel drug delivery approach. The insulin results are interesting because they provide proof of principle for the approach but much remains to be done to demonstrate that this approach will be an effective way to treat diabetes, however. This aspect of the discussion should be tempered.

Responses: Thank you for your advice.

We have supplemented the following discussions in our revised manuscript.

“Additionally, the hepatoselective insulin analogues could elicit overly hepatic insulinization and lead to abnormal hepatic lipid accumulation which might increase the risk of liver steatosis^{7, 8}. In comparison, the Pep/Gal-PNPs could mimic the biodistribution of endogenous insulin by actively targeting insulin to the liver, simultaneously maintaining the optimal insulinization in the liver and the periphery. Therefore, Pep/Gal-PNPs show advantageous for effective diabetes treatment and reduce the risk of adverse effects compared with hepatoselective insulin analogues⁹.”
(Page 18, Line 554-561)

References

1. Xia, Q., Ding, H. & Ma, Y. Can dual-ligand targeting enhance cellular uptake of nanoparticles? *Nanoscale* **9**, 8982-8989 (2017).
2. Liu, Y. et al. Synergetic combinations of dual-targeting ligands for enhanced in vitro and in vivo tumor targeting. *Adv. Healthcare Mater.* **7**, e1800106 (2018).
3. Guo, P. et al. Dual complementary liposomes inhibit triple-negative breast tumor progression and metastasis. *Sci Adv* **5**, eaav5010 (2019).
4. Kluza, E. et al. Synergistic targeting of alphavbeta3 integrin and galectin-1 with heteromultivalent paramagnetic liposomes for combined MR imaging and treatment of angiogenesis. *Nano Lett.* **10**, 52-58 (2010).
5. Geho, W. B. The importance of the liver in insulin replacement therapy in insulin-deficient diabetes. *Diabetes* **63**, 1445-1447 (2014).
6. Arbit, E. & Kidron, M. Oral insulin delivery in a physiologic context: Review. *J. Diabetes Sci. Technol.* **11**, 825-832 (2017).

7. Edgerton, D. S. et al. Targeting insulin to the liver corrects defects in glucose metabolism caused by peripheral insulin delivery. *JCI Insight* **5**, e126974 (2019).
8. Hirose, T. Development of new basal insulin peglispro (LY2605541) ends in a disappointing result. *Diabetol Int* **7**, 16-17 (2016).
9. Herring, R., Jones, R. H. & Russell-Jones, D. L. Hepatoselectivity and the evolution of insulin. *Diabetes Obes. Metab.* **16**, 1-8 (2014).

REVIEWERS' COMMENTS

Reviewer #1 (Remarks to the Author):

The authors have addressed related issues.